**Subject Category:**
Biology (whole organism)

behaviour/ecology

coping styles, repeatability, glucocorticoids, Middle Atlas, Barbary macaques, state-dependent model

**Author for correspondence:**
P. J. Tkaczynski
e-mail: patrick_tkaczynski@eva.mpg.de,
pjtresearchltd@gmail.com

†Present address: Department of Anthropology, Durham University, Dawson Building, South Road, Durham DH1 3LE, UK.

# Repeatable glucocorticoid expression is associated with behavioural syndromes in males but not females in a wild primate

P. J. Tkaczynski[1,2], C. Ross[2], J. Lehmann[2], M. Mouna[3], B. Majolo[4] and A. MacLarnon[2,†]

[1]Department of Primatology, Max Planck Institute of Evolutionary Anthropology, Deutscher Platz 6, 04103 Leipzig, Germany
[2]Centre for Research in Evolutionary, Social and Inter-Disciplinary Anthropology, University of Roehampton, Holybourne Avenue, London SW15 4JD, UK
[3]Mohammed V University, Institut Scientifique, Rabat, Morocco
[4]School of Psychology, University of Lincoln, Lincoln, UK

PJT, 0000-0003-3207-2132; BM, 0000-0002-0235-3040

Behavioural syndromes are a well-established phenomenon in human and non-human animal behavioural ecology. However, the mechanisms that lead to correlations among behaviours and individual consistency in their expression at the apparent expense of behavioural plasticity remain unclear. The 'state-dependent' hypothesis posits that inter-individual variation in behaviour arises from inter-individual variation in state and that the relative stability of these states within an individual leads to consistency of behaviour. The endocrine stress response, in part mediated by glucocorticoids (GCs), is a proposed behavioural syndrome-associated state as GC levels are linked to an individual's behavioural responses to stressors. In this study, in wild Barbary macaques (*Macaca sylvanus*), consistent inter-individual differences were observed in both sexes for GC activity (faecal glucocorticoid, fGC concentrations), but not GC variation (coefficient of variation in fGC concentrations). The expression of the behavioural syndrome 'Excitability' (characterized by the frequencies of brief affiliation or aggressive interactions) was related to GC activity in males but not in females; more 'excitable' males had lower GC activity. There was no relationship in females between any of the behavioural syndromes and GC activity, nor in either sex with GC variation. The negative relationship between GC activity and Excitability in males provides some support for GC expression as a behavioural syndrome-generating state under the state-dependent framework.

The absence of this relationship in females highlights that state-behavioural syndrome associations may not be generalizable within a species and that broader sex differences in state need to be considered for understanding the emergence and maintenance of behavioural syndromes.

## 1. Introduction

The term 'behavioural syndromes' refers to suites of correlated behaviours that vary consistently between individuals across time and contexts [1–6]. Consistency in behaviour and correlations between different behaviours are potentially maladaptive if this constrains an individual's ability to adapt to a changing environment [7,8]. Therefore, adaptive models have been sought to explain the apparent ubiquity of behavioural syndromes among humans and non-human animals (hereafter animals; [5,9,10]). The 'state-dependent' model posits that inter-individual variation in behavioural repertoires arises from inter-individual variation in 'state' and that the relative stability of states can be related to intra-individual consistency in behaviours and correlations of behaviours [5,9–11]. States can be defined as intrinsic features, such as morphology, health or physiology, or as extrinsic factors, such as habitat or social environment, that affect the cost/benefit trade-offs in strategies related to increasing fitness [10,12].

To date, empirical testing of the state-dependent model has largely focused on hormone levels and metabolic rates as the behaviour-associated states [13]. Theoretically, endocrine functioning can have multi-modal effects on behavioural syndrome expression by impacting on the speed, strength and flexibility of behavioural responses to the immediate environment [14–16]. Studies investigating endocrine factors and behavioural syndromes have often focused on the hormonal stress response via monitoring of cortisol and other glucocorticoids (GCs; [17–20]). In vertebrates, the major response to a stressor is activation of the hypothalamic-pituitary-adrenal (HPA) axis which stimulates the redirection of energy and behaviour via GCs [21–24]. Animals are subjected to stressors constantly, such as challenges to homeostasis in the form of seasonal variation in temperature and food availability, as well as less predictable and potentially more severe stressors, such as predation events or competitive encounters [25–28]. Fluctuations in GC levels are expected to be closely linked with an individual's response to its environment and thus to much of its activity, behaviour and, therefore, potentially correlations of behaviour in the form of behavioural syndromes. Indeed, certain patterns of behaviour have been framed as 'coping styles' to stressors: individuals with 'proactive' coping styles tend to be active and aggressive, and to have lower HPA activity and reactivity than 'reactive' individuals, which tend to use strategies to avoid confrontation [18,19,29].

An assumption of the state-dependent hypothesis is that a behavioural syndrome-associated state, such as GC measures, should be as consistent and repeatable as the behavioural syndrome to which it is associated ([10]; although see [5]). Repeatability refers to the proportion of phenotypic variation that can be explained by consistent individual differences where phenotypic variation is the sum of variance in phenotype expression between and within individuals [30]. Several wild animal studies report low to non-existent levels of repeatability of hormonal stress levels, including GC measures in avian [31,32], fish [17,33], reptile [34], rodent [35] and primate species [36]. Given the degree of within-individual plasticity in expression of circulating hormones, recent reviews have suggested that hormonal reactivity and reactive norms, as opposed to absolute glucocorticoid activity, may be more appropriate measures to relate to other phenotypic traits, such as behavioural syndromes [37,38]. Thus, there are outstanding questions about the repeatability of stress-related hormone expression, whether this is potentially a behavioural syndrome-associated state, and if so, how mechanistic links between state and behavioural syndromes operate. Both the aforementioned state-dependent and coping style frameworks provide non-mutually exclusive paradigms to investigate such mechanisms.

Monitoring of hormonal stress responses in the aforementioned wild animal studies mainly used invasive methods [31–35]. However, trapping and invasive monitoring of animals may affect their glucocorticoid levels and can lead to a sampling bias towards easily trapped individuals [39,40]. This is particularly pertinent when trying to understand adaptive processes and the proximate associations of stress-related hormone measures with behavioural syndromes in natural, non-manipulated environments [2], i.e. in the environments to which species are adapted. Well-established methods allow hormone metabolites to be extracted from faecal or urine samples, and thus the non-invasive monitoring of hormone activity in individuals, including wild subjects [41,42]. Furthermore, previous studies examining associations between behavioural syndromes and GCs have used experimental assays to measure behaviour, for example through exposure to novel objects [31], response to handling

[35] or simulated predator presence [17]. Therefore, to understand potential state-dependent associations between behavioural syndromes and stress physiology, it would be valuable to examine whether stress physiology is related to repeatable behavioural patterns outside of experimental paradigms.

In our study, non-invasive hormone sampling and behavioural syndrome quantification were used to test the state-dependent model by examining whether measures of hormonal stress physiology are behavioural syndrome-associated states in both male and female wild Barbary macaques (*Macaca sylvanus*). This non-human primate is a seasonal breeder [43,44], and faces extreme variation in climate [45,46] as well as a number of anthropogenic pressures [47], making it a useful species for exploring the interplay between consistency in behaviour and GCs in a highly heterogenetic environment. Associations between non-invasively quantified GC levels and behavioural syndromes have previously been studied in wild chacma baboons (*Papio hamadryas ursinus*; [48]). However, that study focused exclusively on female subjects. Sex differences are anticipated in behavioural responses to stress [49,50], thus exploring both sexes in their associations between GCs and behavioural syndromes is necessary to test the state-dependent model more fully, and in turn for understanding how consistent individual differences in behaviour evolve and are maintained.

In our study, we first quantified the repeatability of GC activity (measured by faecal GC [fGC] levels) and GC variation (coefficient of variation in fGC levels). If we observed significant repeatability in GC activity and/ or GC variation, we concluded that the variable(s) met the requirement for the state-dependent model (i.e. consistent inter-individual differences in the expression of the variable). To test the state-dependent model, we then examined whether expression of the repeatable GC variables (GC activity and/or GC variation) were related to the expression of established behavioural syndromes in Barbary macaques. Three independent and non-correlated behavioural syndromes have previously been identified in this species and population [51]: 'Excitability', characterized by the frequencies of brief affiliation or aggressive interactions; 'Sociability', characterized by levels of gregariousness, time spent in the centre of the group and in proximity with conspecifics; and 'Tactility', characterized by proportions of time spent allo- or self-grooming. On a behavioural level, the 'Excitability' syndrome is most structurally similar to the proactive-reactive coping style framework previously described [18,19]. Therefore, we predicted that behaviourally proactive, more excitable individuals would have lower GC activity and lower GC variation when compared to behaviourally reactive, less excitable individuals. The expression of each behavioural syndrome is independent and uncorrelated with that of the other two [51], therefore, we did not expect the same state-dependent relationship between Tactility and Sociability and GC measures, i.e. one state should not be associated with multiple behavioural syndromes. As such, we were unable to make *a priori* predictions for Tactility and Sociability given our predicted relationship between Excitability and GC measures.

# 2. Methods

## 2.1. Study subjects/site

Data were collected from two groups of wild Barbary macaques, Blue and Green groups, located in Ifrane National Park, Morocco (33°24′ N, 05°12′ W; elevation 1500–2000 m.a.s.l.). Both groups were habituated to human researchers and observable from distances of around 7 m. The subjects of this study were the adult males and females of both groups: five males and five females in the Blue group; and seven males, eight females in the Green group. Adults were defined as sexually mature individuals, based on body size for both sexes, plus the presence of anogenital swellings during the breeding season for females, and descended testicles and large canines for males [43]. Two adult females of the Blue group appeared to have experienced reproductive senescence, evidenced by the lack of anogenital swellings and failure to become pregnant during the course of the study. As reproductive state affects GC levels in primates [52–55], these two females were excluded from the analysis.

## 2.2. Behavioural data collection

Behavioural data were used to quantify behavioural syndromes in subjects. These data were collected by PJT and seven research assistants during three time periods: during one mating season (time period 1: October 2013–January 2014) and two post-mating seasons (time period 2: January 2014–March 2014; time period 3: February 2015–April 2015). Each subject was observed for a 30 min focal sample [56] at least three times every 6 days throughout the data collection period. During focal samples, state behaviours (feeding, resting, travelling and grooming) were recorded continuously, and incidents of contact, approaches,

agonistic, dominance and solitary behaviours, as well as facial displays and vocalizations, were recorded as point events (see electronic supplementary material, table S1 for descriptions of behaviours). At the start and end of each focal sample, proximity scans were conducted to record the number of group members within 0–1 m, 1–5 m and 5–10 m of the focal subject. Additionally, observers recorded whether subjects were central or peripheral within the group in terms of spatial position; individuals were considered peripheral if they were the outermost individual at the front, rear or side of the group, and otherwise were considered to be central. Ad libitum observations [56] of all aggressions, submissions and supplants were recorded to inform analyses of dominance hierarchies.

All behavioural data were collected using a Psion handheld computer and The Observer XT software v.8.0 (Noldus Information Technology, 2008). Inter-observer reliability tests using intra-class coefficients (ICC; [57]) were conducted. Research assistants started collecting behavioural data only once they had recorded two consecutive tests where the frequency and duration of variables recorded were significantly correlated with those recorded by PJT (ICC > 0.95; $p < 0.05$).

## 2.3. Weather data collection

In the first two time periods, air temperature was recorded hourly when in the presence of subjects using a 3500 Kestrel Pocket Weather Station. In the third time period, air temperature was recorded only when the subjects left or entered sleeping trees in the morning and evening, respectively. In all time periods, temperatures were recorded in the shade and with the device 1.5 m above the ground [46]. In total, 111 temperature recordings were recorded during the entire study period, an average of 13.75 (±6.43) temperature recordings per study month and 28, 40 and 42 temperature recordings for time periods 1, 2 and 3, respectively.

## 2.4. Faecal sample collection

Faecal samples were collected opportunistically throughout the study period aiming for at least one sample per subject every 6 days. In total, 816 faecal samples were collected and analysed for fGC concentration, averaging 32.14 ± 2.76 samples per subject and 10.90 ± 0.86 per subject per study time period (table 1). Samples were collected within 15 min of defecation: the faeces were first homogenized and a 3–5 g sample portioned into a 30 ml Azlon tube (Azlon 7BWH0030N, Azlon, Stone, Staffordshire, UK), which in turn was placed in an ice bag and kept cold before being transferred to a freezer (−20°C) at the end of each field day. Faeces contaminated with urine or that appeared diarrheal were not sampled. Samples were transported in cool boxes packed with ice to the University of Roehampton at the end of each field period, following receipt of an attestation of health for the subjects from Moroccan authorities and under DEFRA import licences.

## 2.5. Quantifying faecal glucocorticoids and stress variables

Faecal samples were freeze-dried (Edwards Freeze Dryer Modulyo EF4), pulverized using a pestle and mortar, and any undigested material (seeds, nuts etc.) removed. Single extraction [58] was used: steroid hormones were extracted from 50–90 mg of powdered faecal matter using validated and published procedures [59,60]. Radio-recovery testing using radio-labelled oestradiol established an average 85.03% recovery rate from 12 control samples using this steroid extraction method for samples from the same species and site [61].

fGC concentrations were measured using a competitive binding enzyme-immunoassay for 5 $\beta$ -androstane-3$\alpha$, 11 $\beta$ -diol-17-one, previously validated for the measurement of GC metabolites in Barbary macaques [60], as well as in long-tailed (*Macaca fascicularis*) [60], rhesus (*M. mulatta*) and Assamese macaques (*M. assamensis*) [62], chimpanzees (*Pan troglodytes*) and lowland gorillas (*Gorilla gorilla*) [60]. Mean intra-assay coefficients of variation were 3.68% for high ($n = 16$) and 6.51% for low ($n = 17$) concentrations. Inter-assay coefficients of variation were 7.00% for high and 11.67% for low concentrations ($n = 33$ plates). Assay sensitivity measured for 90% binding was 1.83 pg/50 µl.

GC activity was quantified using individual fGC concentrations (ng g$^{-1}$ dry faecal weight). GC variation for each subject animal for each of the three time periods was determined using a coefficient of variation for fGC concentrations, fGCcv [63] corrected for the number of samples included in each calculation:

$$fGCcv = (1 + \frac{1}{4n}) \times \frac{\text{standard deviation fGC}}{\text{mean fGC}} \times 100.$$

Table 1 provides descriptive statistics for each of the GC measures.

**Table 1.** Summary statistics for response (GC measures) variables included in models. There were 25 subjects (12 male; 13 female) included in the study. All variables were to achieve normality prior to inclusion in models: square root transformed for GC activity, $\log_{10}$ transformed for GC variation.

| model | variable | sample size (mean per subject $\pm$s.d.) | | values: mean ($\pm$s.d.) | |
|---|---|---|---|---|---|
| | | males | females | males | females |
| long-term repeatability of GC activity | faecal glucocorticoid concentration (ng g$^{-1}$ dry faecal weight)[a] | 391 (33 $\pm$ 3) | 426 (33 $\pm$ 3) | 1302 ($\pm$505) | 1663 ($\pm$765) |
| short-term repeatability of GC activity: mating season 2013–2014 | faecal glucocorticoid concentration (ng g$^{-1}$ dry faecal weight)[b] | 149 (12 $\pm$ 3) | 167 (13 $\pm$ 3) | 1271 ($\pm$243) | 1523 ($\pm$403) |
| short-term repeatability of GC activity: post-mating season 2014 | | 93 (8 $\pm$ 1) | 99 (8 $\pm$ 1) | 1403 ($\pm$188) | 1695 ($\pm$543) |
| short-term repeatability of GC activity: post-mating season 2015 | | 149 (12 $\pm$ 2) | 160 (12 $\pm$ 1) | 1266 ($\pm$166) | 1796 ($\pm$428) |
| repeatability of GC variation | fGCcv (coefficient of variation in faecal glucocorticoid concentrations; %)[c] | 36 (3 $\pm$ 0) | 45 ($\pm$0) | 132 ($\pm$23.2) | 138 ($\pm$21.54) |

[a]All faecal sample values across all time periods.
[b]All faecal sample values within each time period.
[c]One value per subject per time period.

## 2.6. Statistical analysis

All statistical analyses were performed in R v.3.2.3 [64].

### 2.6.1. Quantifying behavioural syndromes

Three independent, non-correlated and consistently expressed behavioural syndromes have previously been identified in our study subjects: Excitability, Sociability and Tactility (electronic supplementary material, table S1; [51]). The scores for each of these behavioural syndromes were taken from a former study [51] for use in our present analysis. Quantification of these behavioural syndrome scores involved two stages: first, testing for repeatability in the expression of behavioural variables from focal and scan observations; second, examining correlations among behavioural variables found to be repeatable [45]. Repeatability was tested using a linear mixed-effect model (LMM) estimate of repeatability, $R_A$. This repeatability coefficient is based on the degree of within-individual variation in the expression of a variable compared to the overall variation between-individuals in expression of the variable. Significance of the coefficient is based on a randomization procedure whereby variables are

randomized between factors (subjects) without replacement 10 000 times and $R_A$ calculated for each randomization of the data; the $p$-value is calculated as the proportion of simulations that results in a random $R_A$ greater than or equal to the observed $R_A$ [65]. The repeatability analysis was calculated using the *rptr* package in R [66]. Of the 28 behavioural variables tested, 18 were significantly repeatable [51]. Factor analyses, performed using the *psych* package [67] in R, then identified correlations among these repeatable variables. The sum of the mean values of the behavioural variables that loaded onto a particular factor was used to create individual (subject-specific) scores for a particular behavioural syndrome (electronic supplementary material, table S1). One score per behavioural syndrome per time period was allocated to each subject, with higher scores indicating greater expression of the syndrome [45].

### 2.6.2. Quantifying rank

Rank calculations were based on 1236 dyadic agonistic interactions between subjects observed between 9 October 2013 and 18 April 2015 during focal samples and ad libitum observations [56]. Calculations were performed using an Elo-rating procedure [68] and the *EloRating* package in R [69]. For each subject, rank was extracted at the end of each month (each 30 days from the start of the study) and at the end of each data collection time block. The starting point for each Elo-rating extraction and calculation was the commencement of data collection (i.e. 9 October 2013). Dominance hierarchies were analysed separately for the two groups, and for males and females, and Elo rankings for each month and time period were standardized for each group-sex category to a mean of 0 and a standard deviation of 1 prior to inclusion in statistical models (table 1).

### 2.6.3. GC expression as a potential state under the state-dependent model: repeatability of GC activity and variation

Repeatability of GC activity (individual fGC concentrations; ng g$^{-1}$ of dry faecal weight) and of GC variation (fGCcv; seasonal coefficients of variation) across time and context was tested using the aforementioned LMM-based estimate of repeatability, $R_A$, using the *rptr* package in R [66]. A series of LMMs were constructed to investigate the longer- and shorter-term repeatability of GC activity, across the whole study period and within individual time periods, respectively, in male and in female subjects. Table 1 lists the dependent variables and their summary statistics for these models.

For measuring longer-term repeatability, for each sex, a model was constructed with the fGC concentrations of all faecal samples across all time periods as the dependent variable (square root transformed in order to reach normality). To measure shorter-term repeatability, models for each sex and each time period were constructed (i.e. six separate models).

Following Sonnweber *et al.* [36], control variables were included in the models to render repeatability estimates independent of social, environmental or physiological factors known to affect fGC concentrations in Barbary macaques. Specifically, we controlled for any potential effect of dominance rank [70–72] and mean minimum daily temperatures [73] via the inclusion of monthly Elo scores (standardized to mean of 0 and standard deviation of 1) and monthly mean minimum daily temperature (standardized to mean of 0 and standard deviation of 1) in both male and female models. All female subjects gave birth following both post-mating seasons in 2014 and 2015; however, we do not have precise birth date data for all females to be able to distinguish between early- or late-stage pregnancy. Therefore, we treated all females as pregnant during time periods 2 and 3; thus, the time period categorization controlled for reproductive state in our models [52–55]. For both male and female models, time period was also included to control for other seasonal and temporal factors which could reduce robustness of repeatability measures [74]. Group identity was also included in both male and female models as a fixed effect. In each model, subject identity was included as a random effect in order to generate the repeatability estimates.

Two LMMs were also constructed to investigate the longer-term repeatability of GC variation: one for male subjects and one for female subjects. The dependent variable was fGCcv values for each subject in each time period (i.e. three per subject), log$_{10}$ transformed in order to achieve normality for analysis. Both the male and female models included time period (which incorporates reproductive state for females), group, standardized Elo scores (score extracted on the last day of each time period) and the mean of minimum monthly temperatures within a time period as control variables, with subject identity included as a random effect.

For all models, collinearity of fixed factors was examined using variance inflation factors (VIF) and the *fsmb* R package [75]. For the GC variation models, collinearity between time period and mean minimum monthly temperatures was observed; therefore, the time period variable was retained. For all other models, no issues with collinearity were observed (all VIFs less than 2.000). Assumptions of normally distributed and homogeneous residuals were verified by visual inspection of qq-plots and residuals plotted against fitted values, with no apparent issues discernible. We also assessed model stability removing one level of each random effect at a time and recalculating the estimates of the different predictors; this revealed no stability issue. To determine if any of the included control variables significantly affected the GC variables and thus the repeatability of the variables, the long-term repeatability models were analysed using a full model approach [76]. All models were fitted for analysis using the R package *lme4* [77].

### 2.6.4. Associations between behavioural syndromes and GC activity/variation

Where long-term repeatability was identified for either GC measure (GC activity or GC variation), LMMs were used to examine the relationships between the expression of behavioural syndromes and the GC measure. For these models, the dependent variable was the behavioural syndrome score for each time period (i.e. one model per behavioural syndrome, three scores per subject per behavioural syndrome). The predictor variable was the random effect estimate (best linear unbiased predictor; [78]) for each subject derived from the GC repeatability models. These estimates are the GC phenotype for each subject, with each estimate representing the average GC value (activity or variation) for a subject given the fixed effects of rank, group and minimum monthly temperature on GC values within a particular time period, making three data points per subject ($n = 75$). In all models, subjects were random effects to control for pseudoreplication of multiple behavioural syndrome scores from individual subjects. As the sexes varied significantly in the expression of certain behavioural syndromes [51], separate models were run for males and females.

Where any association between a behavioural syndrome and either GC variable was identified, we performed a *post hoc* analysis to examine if any individual behavioural variable might be driving the association between any particular behavioural syndrome and either GC activity or variation. To do so, we constructed a series of LMMs with the mean hourly rate of expression for an individual behavioural variable constituting the behavioural syndrome [51] as the dependent variable (one value per subject per time period, $n = 75$) and the same GC phenotype variable used in testing the association between behavioural syndrome scores and the GC variables as the predictor variable. Once again, subject identity was included as a random effect to control for pseudoreplication of multiple behavioural syndrome scores from individual subjects.

All models were again fitted using the R package *lme4* [77] and analysed using a full model approach [76]. The models were again visually assessed to see they conformed to model assumptions and examined for stability, revealing no clear issues.

## 3. Results

### 3.1. GC expression as a potential state under the state-dependent model

Consistent inter-individual differences in GC activity (fGC concentration [ng g$^{-1}$]) were observed between male ($R_A = 0.078$; $p < 0.001$; table 2 and figure 1*a*) and between female ($R_A = 0.063$; $p < 0.001$; table 2 and figure 1*b*) subjects across the three time periods combined. For both males and females, short-term repeatability was only significant during mating season 2013–2014 (male $R_A = 0.206$; $p < 0.001$; female $R_A = 0.171$; $p < 0.001$; table 2). There was no evidence for consistent inter-individual differences in GC variation (fGCcv values) between males ($R_A < 0.001$; $p = 0.629$; table 2 and figure 1*c*) nor between females ($R_A = 0.001$; $p = 0.348$; table 2 and figure 1*d*).

Male GC activity was higher in both the post-mating time periods compared with the mating time period. Compared with the mating season in 2013–2014, we observed significantly higher levels of GC activity in the post-mating season in 2014 (LMM; $p < 0.001$; table 3*a*) and the post-mating season in 2015 (LMM; $p = 0.026$; table 3*a*). In females, GC activity was non-significantly higher in the post-mating season in 2014 compared with the mating season in 2013–2014 (LMM; $p = 0.428$); table 3*b*); however, GC activity was significantly higher in the post-mating season in 2015 compared with the mating season in 2013–2014 (LMM; $p < 0.001$; table 3*b*). For females, individuals from the Green group

**Table 2.** Repeatability estimates ($R_A$) for male and female GC activity (fGC concentrations [ng g$^{-1}$; square root transformed]) and GC variation (coefficient of variation in fGC concentrations; log$_{10}$ transformed). Significant ($p < 0.05$) repeatability estimates are in italics. For GC activity, $R_A$ values were calculated across the whole study ($n = 391$ faecal samples for males; $n = 426$ faecal samples for females) and within each of the three time periods of the study (table 1 for sample sizes within time periods for the two sexes). For GC variation, $R_A$ values were calculated across the whole study only ($n = 36$ coefficients for males, one per time period [$n = 3$] per subject [$n = 12$]; $n = 39$ coefficients for females, one per time period [$n = 3$] per subject [$n = 13$]).

| terms | $R_A$ (s.e.) | 95% confidence intervals | *p*-value |
|---|---|---|---|
| GC activity | | | |
| males | | | |
| across whole study period | *0.078 (0.045)* | *(0.004, 0.178)* | *<0.001* |
| within mating season 2013–2014 | *0.206 (0.094)* | *(0.030, 0.395)* | *<0.001* |
| within post-mating season 2014 | <0.001 (0.043) | (<0.001, 0.146) | >0.999 |
| within post-mating season 2015 | 0.049 (0.050) | (<0.001, 0.172) | 0.113 |
| females | | | |
| across whole study period | *0.063 (0.037)* | *(0.002, 0.146)* | *<0.001* |
| within mating season 2013–2014 | *0.205 (0.089)* | *(0.034, 0.397)* | *<0.001* |
| within post-mating season 2014 | <0.001 (0.042) | (<0.001, 0.140) | >0.999 |
| within post-mating season 2015 | 0.059 (0.057) | (<0.001, 0.196) | 0.113 |
| GC variation | | | |
| males | <0.001 (0.137) | (<0.001, 0.450) | 0.629 |
| females | 0.119 (0.161) | (<0.001, 0.534) | 0.348 |

had significantly lower GC activity than those of the Blue group (LMM; $p < 0.001$; table 3*b*). In males, we also observed a trend for individuals from the Green group to have lower GC activity than individuals from the Blue group (LMM; $p = 0.086$; table 3*a*).

Male GC variation in the post-mating season in 2014 was significantly lower than in mating season 2013–2014 (LMM; $p < 0.001$; table 3*c*) and there was a trend for GC variation to be different between the mating season 2013–2014 and post-mating season in 2015 (LMM; $p = 0.068$; table 3*c*). As with males, female GC variation was higher in the mating season 2013–2014 than the post-mating season 2014 (LMM; $p = 0.049$; table 3*d*); however, no differences were observed between mating season 2013–2014 and post-mating season 2015.

## 3.2. Associations between behavioural syndromes and GC activity/variation

In males, higher Excitability scores were associated with lower GC activity (LMM; $p = 0.047$; table 4*a* and figure 2). A *post hoc* analysis found no association between the majority of the constituent behavioural variables of Excitability (yawns, tree shakes, mounts, contact aggression, open mouths, genital touches and triadic embraces; see [51] for definitions and details) and GC activity in male subjects (table 5*a–e*). However, a significant positive relationship was observed between the frequency of embraces and GC activity (LMM; $p = 0.019$; table 5*f*). Tactility and Sociability scores were not correlated with GC activity in males (table 4*b,c*); in females, there were no significant relationships observed between GC activity and the expression of behavioural syndromes (table 4*d–f*).

As repeatability in GC variation was not observed in either sex, the relationship between this GC variable and behavioural syndrome scores was not examined.

## 4. Discussion

Our study examined GC expression as a potential behavioural syndrome-associated state under the framework of the state-dependent model employed to explain consistent inter-individual differences in behavioural repertoires. In both male and female Barbary macaques, we observed consistent individual differences in GC activity but not GC variation. In males, but not females, the expression

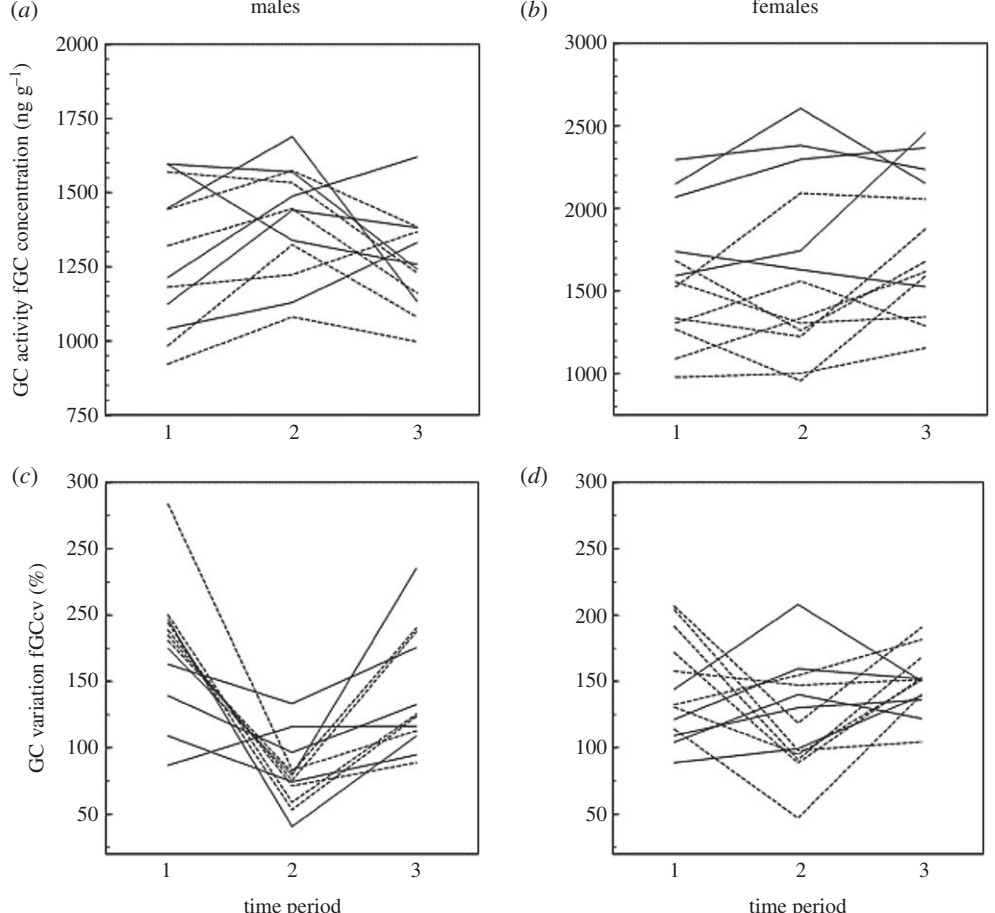

**Figure 1.** Consistency in GC activity (fGC concentration; ng g$^{-1}$) in male (*a*) and female (*b*) subjects; and consistency in GC variation (fGCcv) in male (*c*) and female (*d*) subjects. Each line represents an individual and the change in their mean values for GC activity or variation across the three time periods (1 = mating season 2013–2014; 2 = post-mating season 2014; 3 = post-mating season 2015). Dashed lines represent subjects from the Green group (*n* = 7 for male plots; *n* = 8 for female plots), solid lines subjects from the Blue group (*n* = 5 for male plots; *n* = 5 for female plots).

of Excitability was associated with levels of GC activity, highlighting sex differences in the association between GC physiology and the expression of behavioural syndromes. Sociability and Tactility were not associated with GC activity or variation in either sex.

## 4.1. Repeatability of GC variables

Although statistically significant, the longer-term repeatability estimates (across all time periods) for Barbary macaque male and female GC activity were low (less than or equal to 0.10). This finding is consistent with other wild animal studies that report low levels of intra-individual consistency over time and context in physiological stress levels (reviewed in [15,30]). It has been proposed that this inconsistency might be a consequence of the highly heterogenic environment wild animals occupy in terms of climate, food availability and social interactions, which generates considerable variation in the experience of stressors within and between individuals [15]. In our repeatability models, we controlled for environmental (minimum temperatures) and social factors (rank, group), yet still found low levels of repeatability. For both sexes, the highest repeatability estimates for GC activity were found during time period 1, a mating season, and thus a period when males are aggressively competing and forming coalitions to gain access to females [44], and females are probably experiencing increased harassment from males [79]. Although GC activity was not higher during this period compared to other time periods in either sex, this elevated intra- and inter-sex harassment and aggression may result in the clearest consistent individual differences between subjects in GC activity as individuals respond variably both physiologically and behaviourally to

**Table 3.** Models describing variation in GC activity (fGC concentrations [ng g$^{-1}$; square root transformed]) among (*a*) male (*n* = 391 faecal samples) and (*b*) female (*n* = 426 faecal samples) subjects; and models describing variation in GC variation (coefficient of variation in fGC concentrations; log$_{10}$ transformed) among (*c*) male (*n* = 36 coefficients) and (*d*) female (*n* = 45 coefficients) subjects. Significant (*p* < 0.05) terms are in italics. For both models, for the group categorical variable, the Green group is the reference category, while for the time period categorical variable, time period 1 (mating season 2013–2014) is the reference category.

| terms | estimate (s.e.) | 95% confidence intervals | *t*-value | *p*-value |
|---|---|---|---|---|
| (*a*) male GC activity model | | | | |
| (intercept) | 35.053 (1.004) | (32.961, 37.120) | 34.925 | |
| time period: mating season 2013–2014 | | | | |
| post-mating season 2014 | *3.558 (0.873)* | *(1.841, 5.274)* | *4.074* | *<0.001* |
| post-mating season 2015 | *1.735 (0.778)* | *(0.205, 3.262)* | *2.231* | *0.026* |
| group | −1.951 (1.138) | (−4.383, 0.501) | −1.715 | 0.086 |
| rank | −0.253 (0.418) | (−1.181, 0.601) | −0.606 | 0.545 |
| mean minimum temp | *2.466 (0.339)* | *(1.800, 3.132)* | *7.275* | *<0.001* |
| (*b*) female GC activity model | | | | |
| (intercept) | 43.737 (1.198) | (41.294, 46.207) | 36.505 | |
| time period: mating season 2013–2014 | | | | |
| post-mating season 2014 | 0.841 (1.063) | (−1.246, 2.930) | 0.792 | 0.428 |
| post-mating season 2015 | *2.874 (0.935)* | *(1.038, 4.711)* | *2.874* | *0.002* |
| group | *−8.227 (1.335)* | *(−11.049, −5.422)* | *−6.161* | *<0.001* |
| rank | −0.087 (0.455) | (−0.983, 0.813) | −0.192 | 0.848 |
| mean minimum temp | 0.711 (0.412) | (−0.098, 1.521) | −6.161 | 0.084 |
| (*c*) male GC variation model | | | | |
| (intercept) | 2.232 (0.045) | 2.142 2.323 | 49.597 | |
| time period: mating season 2013–2014 | | | | |
| post-mating season 2014 | *−0.341 (0.022)* | *(−0.447, −0.235)* | *−6.490* | *<0.001* |
| post-mating season 2015 | −0.096 (0.053) | (−0.202, 0.010) | −1.826 | 0.068 |
| group | −0.011 (0.053) | (−0.101, 0.078) | −0.258 | 0.797 |
| rank | −0.014 (0.044) | (−0.059, 0.032) | −0.649 | 0.516 |
| (*d*) female GC variation model | | | | |
| (intercept) | 2.156 (0.041) | 2.072 2.239 | 52.476 | |
| time period: mating season 2013–2014 | | | | |
| post-mating season 2014 | *−0.086 (0.044)* | *(−0.175, 0.003)* | *−1.966* | *0.049* |
| post-mating season 2015 | 0.025 (0.044) | (−0.064, 0.114) | 0.570 | 0.569 |
| group | −0.019 (0.041) | (−0.106, 0.069) | −0.453 | 0.651 |
| rank | −0.010 (0.018) | (−0.047, 0.026) | −0.543 | 0.587 |

these social stressors. Indeed, repeatability coefficients in the other time periods were low and non-significant for both sexes, suggesting the higher repeatability estimates observed in the mating season may have driven the significance of the long-term repeatability estimates. This result has implications for the time frame in which future research calculates repeatability for hormonal stress levels. Such calculations should cover a range of biologically meaningful time periods for the chosen study species (such as mating and post-mating season in our study). Should the study not encompass all such periods or focus on a period when inter-individual variation in the experience of stressors is low, repeatability estimates may not reflect the general pattern of repeatability for GC levels in that species or population.

**Table 4.** Models describing relationship between behavioural syndrome scores and GC activity (random effect estimate from repeatability models) in Barbary macaques. Models (a), (b) and (c) describe relationship between male Excitability, Sociability and Tactility scores and GC activity ($n = 36$ scores per model; one per subject per time period). Models (d), (e) and (f) describe relationship between female Excitability, Sociability and Tactility scores and GC activity ($n = 45$ scores per model; one per subject per time period). Significant ($p < 0.05$) terms are in italics.

| terms | estimate (s.e.) | 95% confidence intervals | t-value | p-value |
|---|---|---|---|---|
| (a) male Excitability model | | | | |
| (intercept) | 2.532 (0.231) | (2.041, 3.022) | 10.982 | |
| GC activity estimate | −0.235 (0.118) | (−0.474, 0.006) | −1.986 | 0.047 |
| (b) male Sociability model | | | | |
| (intercept) | 1.177 (0.082) | (1.002, 1.352) | 14.324 | |
| GC activity estimate | −0.037 (0.036) | (−0.109, 0.038) | −1.040 | 0.299 |
| (c) male Tactility model | | | | |
| (intercept) | 0.463 (0.030) | (0.398, 0.527) | 15.222 | |
| GC activity estimate | 0.035 (0.021) | (−0.008, 0.077) | 1.643 | 0.100 |
| (d) female Excitability model | | | | |
| (intercept) | 0.863 (0.095) | (0.663, 1.063) | 9.130 | |
| GC activity estimate | −0.034 (0.063) | (−0.161, 0.094) | −0.532 | 0.595 |
| (e) female Sociability model | | | | |
| (intercept) | 1.198 (0.077) | (1.036, 1.360) | 15.623 | |
| GC activity estimate | 0.058 (0.033) | (−0.012, 0.125) | 1.723 | 0.085 |
| (f) female Tactility model | | | | |
| (intercept) | 0.792 (0.032) | (0.725, 0.859) | 24.962 | |
| GC activity estimate | 0.006 (0.020) | (−0.035, 0.047) | 0.310 | 0.757 |

During the post-mating seasons in 2014 and 2015 (time periods 2 and 3), male–male competition was probably diminished, which may have lowered inter-individual variation in the experience of social and metabolic stressors associated with this competition. In both time periods 2 and 3, all female subjects were pregnant (evidenced by the subsequent births observed). Pregnancy, especially late-stage pregnancy, is associated with peaks in cortisol levels in primates [52,54,55,80–83], and in our study, the highest GC concentrations were observed when females were pregnant. Although significant differences in GC activity were observed between the different time periods for males, the female longer-term repeatability estimate for GC activity was lower than in males, which may be related to the variation experienced by females related to pregnancy. Barbary macaques are seasonal breeders, and adult females typically experience one pregnancy *per annum* [43], thus these patterns of increasing cortisol levels during pregnancy may result in low levels of overall GC activity repeatability, with implications for how consistent behavioural patterns may be associated with GC activity in females (discussed below).

While the repeatability of GC activity and/or baseline levels of GCs is relatively well explored, there is little empirical data on the repeatability of stress reactivity, particularly in a natural setting [36]. In our study, using fGCs constrained our ability to assess the repeatability of stress reactivity using reaction norm approaches, i.e. whether consistent inter-individual differences exist in the degree to which individuals physiologically respond to a specific stressor [30]. As fGCs are excreted at variable rates and are a cumulative measure of GC secretion [41,84–86], associating a sample with an individual event, such as an aggression or inter-group conflict, is challenging. Instead, we used a measure representing the overall variation in physiological stress an individual experienced during individual time periods. We found no evidence of consistent individual differences in either sex for this measure of GC variation. As Barbary macaques live in a highly heterogenic environment, meaningful measures of repeatability in GC variation across multiple meteorological and social gradients may be difficult to identify, particularly if conditions in the wild also limit the sampling frequency. Indeed, visual examination of the plots (figure 1c,d) and the data suggest that the low levels of repeatability are

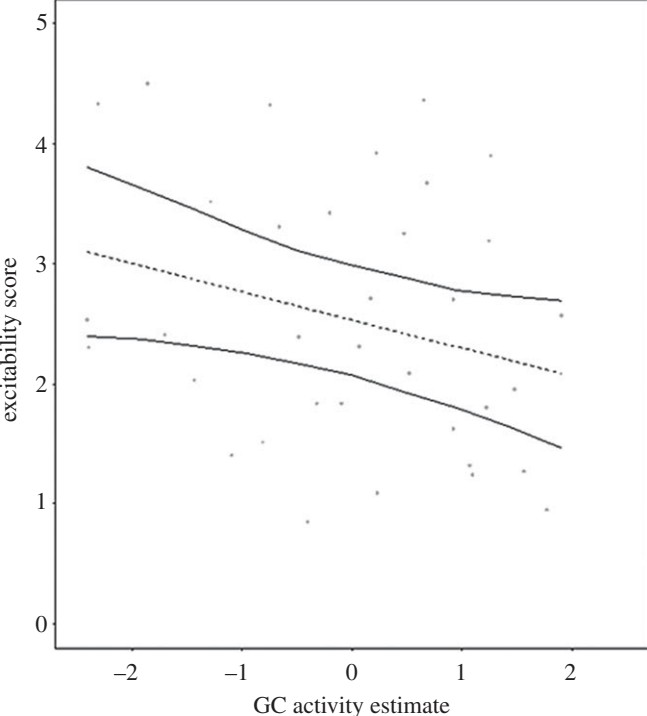

**Figure 2.** The predicted relationship between Excitability scores and GC activity (random effect estimate from repeatability models) in male Barbary macaques ($n = 36$ estimates; 1 per subject per time period). The dashed line represents the relationship by the model (table 4a), the solid lines represent upper and lower 95% confidence intervals.

attributable to low inter-individual variance in GC variation within each of the three time periods. Recent studies in wild animals have used non-invasive methods to monitor specific hormonal responses to a standardized social interaction or stressor, such as single events of affiliation or aggression [87,88]. Such approaches may generate more distinct differences between individuals in terms of reactivity compared to our use of an overall coefficient of variation.

## 4.2. Behavioural syndromes and GC expression

In male Barbary macaques, individuals with higher Excitability typically had lower levels of GC activity. As such, in male Barbary macaques, Excitability is structurally consistent with the proactive-reactive framework of coping styles proposed by Koolhaas *et al.* [19]: individuals with higher Excitability scores were behaviourally and physiologically 'proactive' (tending to be active and aggressive, with lower HPA activity), while individuals with lower Excitability scores were behaviourally and physiologically 'reactive' (avoiding social interactions with higher HPA activity). This finding is significant as associations between behavioural syndromes and GC levels have typically been identified using experimental approaches or invasive hormone monitoring. Indeed, coping styles are typically determined by the immediate responses (both physiological and behavioural) of individuals to an introduced stressor [18]. Our results suggest that proactive-reactive phenotypes, both behaviourally and physiologically, may be apparent in non-manipulated settings and individuals. A useful investigation of this would be to examine the more immediate behavioural and physiological responses of individuals to a stressor using the methods previously proposed to determine the repeatability of reaction norm measures [87,88]. Determining whether the physiological and behavioural associations of Excitability in males are related to these immediate responses to a stressor would reveal the extent to which traditional 'coping style' behaviour and physiology relates to longer-term behavioural strategies and physiological state.

A recent study in captive male Barbary macaques found no association between fGCs and two forms of aggression: proactive, in which aggression is performed to achieve a specific goal, such as resource acquisition; and reactive, in which aggression is performed in response to threats or aggression received [89]. Rates of aggression are a component of Excitability [51], and rates of affiliation are also correlated within this behavioural syndrome, i.e. highly excitable individuals aggress more frequently,

**Table 5.** *Post hoc* analyses examining the relationship between Excitability behavioural variables (see [51] for definitions) and GC activity (random effect estimate from repeatability models) among male subjects ($n = 36$ estimates, 1 per subject per time period). Significant ($p < 0.05$) terms are in italics.

| terms | estimate (s.e.) | 95% confidence intervals | t-value | p-value |
|---|---|---|---|---|
| (a) yawn model | | | | |
| (intercept) | 0.788 (0.079) | (0.619, 0.957) | 9.918 | |
| GC activity estimate | −0.101 (0.057) | (−0.216, 0.013) | −1.779 | 0.075 |
| (b) tree shake model | | | | |
| (intercept) | 0.162 (0.040) | (0.619, 0.957) | 4.047 | |
| GC activity estimate | −0.025 (0.020) | (−0.216, 0.013) | −1.223 | 0.221 |
| (c) mount model | | | | |
| (intercept) | 0.149 (0.034) | (0.080, 0.217) | 4.409 | |
| GC activity estimate | 0.020 (0.028) | (−0.036, 0.076) | 0.713 | 0.476 |
| (d) contact aggression model | | | | |
| (intercept) | 0.095 (0.024) | (0.043, 0.147) | 3.920 | |
| GC activity estimate | 0.007 (0.007) | (−0.008, 0.023) | 1.007 | 0.314 |
| (e) open mouth model | | | | |
| (intercept) | 0.661 (0.073) | (0.505, 0.817) | 9.028 | |
| GC activity estimate | −0.041 (0.051) | (−0.143, 0.062) | −0.804 | 0.421 |
| (f) genital touch model | | | | |
| (intercept) | 0.029 (0.008) | (0.012, 0.045) | 3.675 | |
| GC activity estimate | 0.003 (0.005) | (−0.007, 0.013) | 0.594 | 0.553 |
| (g) triadic embrace model | | | | |
| (intercept) | 0.374 (0.055) | (0.257, 0.491) | 6.811 | |
| GC activity estimate | 0.012 (0.042) | (−0.074, 0.097) | 0.298 | 0.766 |
| (h) embrace model | | | | |
| (intercept) | 0.286 (0.055) | (0.169, 0.403) | 5.214 | |
| *GC activity estimate* | *0.065 (0.028)* | *(0.009, 0.123)* | *2.344* | *0.019* |

but also exchange affiliation more frequently as well. Our *post hoc* analyses did not reveal any clear relationships between the majority of individual behaviours that constitute Excitability and GC activity; the only significant relationship identified was a tendency for individuals which had higher frequencies of embraces also having higher GC activity levels. This positive relationship between embraces and GC activity was in the opposite direction of the general negative relationship between Excitability and GC activity. According to the state-dependent model, considering individual behaviours, as performed in our *post hoc* analysis and the aforementioned study of aggression in Barbary macaques [89], is unlikely to reveal associations between said individual behaviours and physiological states, such as GC phenotypes, as the model proposes that state links a number of behavioural tendencies [5,74]. Our study results support the assertion that certain states link a number of behaviours to generate a broad behavioural phenotype [5,9–11], rather than a particular state predicting the frequency of individual behaviours *per se*.

Understanding how correlations between behaviours within behavioural syndromes arise and are maintained is an ongoing topic in behavioural ecology [5,9,90], and our results for male Barbary macaques provide some support with the state-dependent model [5,9–11]. However, one key caveat to our results and study, in general, is that our findings are purely correlational and we were unable to identify the causal direction of the relationship between GC activity (state) and Excitability (behavioural syndrome). While our results are coherent with the predictions of the state-dependent model, it is equally possible that the consistent inter-individual differences in behaviour may be derived from another untested state, which might, in turn, result in consistent inter-individual

differences in GC levels in males. To address this, researchers could study multiple states concurrently or single states across longer time spans and different life-history stages [5,11,91]. Indeed, a recent meta-analysis of state-dependent studies concluded limited evidence for the model [92], which may be attributable to inappropriate empirical testing of the model, i.e. studying one state in isolation rather than the interactions between multiple states. In the presented study, our sample size and sampling frequency of individuals was limited, while the time frame of the study represented a comparatively short period within the lifespan of the study species. Over the course of an individual's life, the relationship between state and behaviour may be dynamic, involve complex feedback loops, and perhaps be nonlinear and/or additive [5,91,92]. An interesting avenue of research would be studying the ontogeny of behavioural syndromes and how they arise and are maintained during development when intrinsic states are highly labile [91]. Using the same approach as in our study, but with immature macaques would aid determining the direction of the relationship between behavioural syndrome and stress physiology should either state or behaviour become consistent within an individual prior to the other.

In Barbary macaques, in both sexes, there are consistent individual differences in the expression of Excitability [51], yet in females, there is no apparent association between their expression and GC expression. Across multiple experimental paradigms in different human and non-human animal studies, sex-differences are evident in behavioural responses to stressors, with males typically demonstrating higher frequencies of behaviours directly in response to either acute or chronic stressors [49,50]. However, Hodes [49] argues that these sex differences often arise due to bias in the experimental set-ups, with certain tests more likely to induce responses in one sex more than the other. Our results suggest that naturally occurring and non-manipulated stressors also reveal sex differences in the association between physiological stress and the manifestation of behaviour which cannot be explained by experimental bias.

Beyond experimental bias, sex-differences in behavioural associations with stress may arise from sex-differences in physiology and sex-specific physiological demands of reproduction [50]. Montiglio *et al.* [35] proposed that female wild chipmunks (*Tamias striatus*) downregulate their physiological stress response during pregnancy to maximize current reproduction by reducing energy used in stress responses. In this study, despite highly seasonal variation in female stress levels, behavioural and hormonal data collected over several years revealed clear associations between cortisol levels and behavioural syndromes in the same female chipmunks [35]. In our study subjects, we found no evidence of such consistent downregulation of stress: females had both higher GC activity and higher GC variation when pregnant (post-mating seasons 2014 and 2015), which may reflect a species difference between chipmunks and macaques in the reproductive costs of pregnancy. Our study also contrasts with the results of Seyfarth *et al.* [48], which identified an association between behavioural syndromes and fGCs in females in a population of wild chacma baboons (*Papio hamadryas ursinus*). In this study, females which scored highly for the 'Loner' phenotype typically had higher fGC concentrations [48]. A key difference between this study and ours was the time scale of the research: Seyfarth *et al.* [48] were able to draw upon 7 years of behavioural and physiological data in their study of baboons, compared to nine months of data in our study. This once again indicates the importance of time frame and scale for future research testing associations between GC expression and behavioural syndromes. Research involving females should incorporate data from several reproductive and non-reproductive periods to see if, on average over these periods, associations between behavioural syndromes and physiological stress are discernible, as was possible in baboons. Nevertheless, sex-specific differences in the association between behavioural syndromes and physiological stress in males and females may still exist and should be explored in other species beyond Barbary macaques. Although variation in GCs is typically used as a proxy measure for variation in stress, these steroid hormones are also involved with immune response [93] and have numerous synergistic and antagonistic relationships with other hormones, including reproductive hormones, the behavioural implications of which remain largely unexplored [94,95]. By incorporating interactions between GCs and female sex hormones, future research may be able to disentangle the proximate causes of sex-specific patterns of associations between GC expression and behaviour.

Our study is one of the first to explore variation between the sexes in the repeatability of GC measures and their associations between GC secretion and behavioural syndromes. Males and females face different challenges and have different proximate goals in relation to fitness. Future research linking these sex-specific GC-behaviour patterns to sex-specific fitness outcomes will help us understand the selective pressures that may be driving these sex differences. To date, we know relatively little about the relationship between GC phenotypes and fitness [96]. Behavioural syndromes may be a

mechanism through which different GC phenotypes maximize fitness. For example, as GC expression can inhibit immune responses, individuals with consistently elevated GCs relative to their conspecifics may be susceptible to long-term injury or infection of wounds arising from aggressive encounters, and thus a strategy of aggression avoidance may improve health and survival outcomes. Indeed, this may be the strategy of less excitable Barbary macaque males that typically have higher GC activity. Exploring associations between GC phenotypes, behavioural syndromes and fitness would significantly advance our understanding of the evolution and maintenance of behavioural syndromes and the repeatability of GC phenotypes from both a proximate and ultimate perspective.

Ethics. All data collection was conducted following ethical approval by University of Roehampton (reference LSC 13/088) and the receipt of research permits 253/2013 and 44/2014 from Haut-Commissariat aux Eaux et Forêts et à la Lutte Contre la Désertification, Royaume du Maroc.

Data accessibility. Data are available from the Figshare data repository: https://figshare.com/articles/macaque_behav_syn_stress/7701701 (doi:10.6084/m9.figshare.7701701) [97]

Authors' contributions. P.J.T. conceived the study; P.J.T., A.M., J.L. and C.R. designed the study. Data were collected by P.J.T. with support from B.M. and M.M. P.T. and A.M. conducted the laboratory analyses. Statistical analyses, figure and manuscript preparation were performed by P.J.T. and all authors contributed to revisions and approved the final manuscript.

Competing interests. The authors have no competing interests.

Funding. This study was funded by the University of Roehampton Vice Chancellor Scholarship awarded to P.J.T. (January 2013).

Acknowledgements. We thank Ifrane National Park, the Haut Commissariat aux Eaux et Forêts et à la Lutte Contre la Désertification, Ecole Nationale Forestiere d'Ingeniuers and Institut Scientifique de Rabat for research permission and facilitation. We thank Melanie LaCava, Kevin Remeuf, Natalie Miller, Marin Deith, Jamie Canepa and Liz Campbell for their contribution to collecting data. We thank Balbir Singh Josen, Mary Mackenzie and Voulla Bergmann for their assistance with the laboratory work. Thank you also to Chris Herridge for his technical assistance managing the data.

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
