## [Reviewer comments · Royal Society Open Science]

Review History

RSOS-190256.R0 (Original submission)

Review form: Reviewer 1

Is the manuscript scientifically sound in its present form?

Yes

Are the interpretations and conclusions justified by the results?

Yes

Is the language acceptable?

Yes

Is it clear how to access all supporting data?

Yes

Do you have any ethical concerns with this paper?

No

Have you any concerns about statistical analyses in this paper?

Yes

Recommendation?

Accept with minor revision (please list in comments)

Comments to the Author(s)

The paper explores very interesting topic and represents significant contribution to the field of animal personality. It is based on solid and intensive data collection. I think the paper is well written and the analysis clearly stated, I particularly appreciate the analysis of repeatability of stress levels measurement.

I have following comments:

37-39 Abstract: The last sentence is not clear to me, do you mean that behavioral syndromes are more associated with reproductive demands in females? You mean that their behavioral type changes based on reproductive stage? Or is it more that, in females, measures of physiological stress might be more associated with reproduction? I think the latter is rather the case.

In the abstract and elsewhere: I do not particularly like using the term “reactivity” for the second measure of stress. It is in fact variation in long-term levels. Traditionally the term reactivity refers to the difference between basal levels and peak levels provoked by stressor (as authors mention in the discussion). This is not what you are measuring in this study (and one can't do it with faecal samples). Although I understand your logic it is just not the correct use of the term and using the term for your measurement could be misleading to the readers. I would use the term variation instead.

Intro line 113: and Discussion. Here you mention one other study that explored stress levels and personality in baboons. This study is however not mentioned in the discussion. Given that it is presented as the only other study on the same topic, I would expect some discussion about its results in comparison to this study (despite that it was only done on females it still should be mentioned)

line 149 and elsewhere: Given that the three study periods have different biological meaning (mating time vs post mating), I think it would be useful to give them names that include the meaning like mating period 1, post mating period 2. And use these in graphs too. Because when reader looks at graphs he/she should know what the time periods refers to (or put it to note)

line 242: Is Elo-rating particularly suitable for the subsequent analyses, or does it have some advantages compared to other more often used measures of rank e.g. rank order, I&IS method, Davids score? I am just curious if there are any advantages to use it in this particularly study.

line 280: specify if group identity was used as fixed or random effect (I assume it was fixed)

line 297: Why are the behavioral syndromes in the model as predictors/independent variables and the hormones as response/dependent variable? In a biological sense I would assume that it should be the other way round: the hormones are affecting the behavioral reaction/type (following same logic as e.g. genes are affecting behavior). I am aware that behavior can affect hormones, but I do not think this is the pattern you are trying to investigate. Also as implied in the introduction (lines 60-61:” endocrine functioning can have multi-modal effects on personality

expression”) following this I expected the hormones to be the independent variable. Is there any reason why authors selected this approach?

figure 2: Is it possible to plot the real data points to the graph too?

Review form: Reviewer 2

Is the manuscript scientifically sound in its present form?

No

Are the interpretations and conclusions justified by the results?

No

Is the language acceptable?

Yes

Is it clear how to access all supporting data?

Yes

Do you have any ethical concerns with this paper?

No

Have you any concerns about statistical analyses in this paper?

Yes

Recommendation?

Major revision is needed (please make suggestions in comments)

Comments to the Author(s)

This paper investigates the link between behavioural syndromes and physiological stress levels (repeated faecal glucocorticoid metabolite measurements) in males and females of two groups of wild Barbary macaques (*Macaca sylvanus*). The main aim of the study is to test the “state-dependent” hypothesis which states that consistent individual differences in behaviour arise from differences in physiological state (e.g. hormones, metabolism). fGC activity (but not reactivity) was repeatable and the authors find evidence for a link between “Excitability” and fGCs in males but not females. Overall, this is an interesting data set collected under natural conditions. I do, however, have concerns/questions regarding the rationale of the conceptual framework(s) and associated statistical analysis which I outline below. I also provide more specific comments/suggestions.

- Line 28: delete “etc.” in brackets

- Line 32/33: delete “a non-human primate species”

- Line 36: replace “; for females there were no....and personality measures” with “but not in females”

- Lines 37-39: The abstract ends on interpreting the non-significant result for females but doesn’t say anything about the result for males and its implications

- Lines 72-75: “certain correlations of behaviour” – this is an inappropriate definition of coping styles (sensu Koolhaas et al., 1999, 2010)
- Lines 77/78: “as the behavioural syndrome concerned” is unclear
- Lines 80-82: please also include examples of studies finding evidence for repeatability in GCs
- Line 88-90: “both...state-dependent and coping style frameworks...” It is not clear why both concepts are used here and there is very little attempt to link these two frameworks and the predictions that can be derived. I think the manuscript would be stronger if the focus was solely on state-dependency (see comments Lines 295 – 317 below).
- Line 96: delete “now”
- Line 129/130: Are the predictions are that straightforward given that predictions for proactive/reactive individuals differ with regards to HPA axis activity/reactivity (see e.g. Koolhaas et al., 1999; Table 3)? I understand that reactivity here means variation and not necessarily high HPA axis activity in response to a stressor. This is likely to cause confusion especially amongst readers who are familiar with the coping style framework/literature.
- Lines 178ff: The description of reproductive states is insufficient/unclear. Why did you not use traditional categories (cycling, pregnant and lactating)? For periods 2 and 3 it should be possible to distinguish early versus late pregnancy (which can result in differences in fGCs, line 383) if birth dates are known. Similarly, if birth dates are known, conceptions could be estimated and females could be classified as cycling/early pregnant. The 4 categories for swellings are not defined and, in my opinion are not useful (and may unnecessarily affect model fitting).
- Lines 183-184: Delete sentence “In total, 2,616.....per subject”
- Lines 187ff: provide sample sizes (move lines 325-326 to methods section)
- Line 193: where samples transported on (dry) ice?
- Line 202: replace “removed” with “collected”
- Lines 202-204: I think this isn't necessary - say extracted following validated and published procedures and add citations?
- Line 207: add other non-human primates for which the assay has been used
- Lines 222ff: it is unclear if the quantification of behavioural syndromes was conducted specifically for the current study or if measures obtained for Reference 45 were “used” in the present study (unfortunately, I cannot access the paper).
- Lines 254-293: the entire section could be much clearer and concise. Also, what is the rationale for investigating repeatability short-and long-term with regards to subsequent analyses (line 259)?
- Lines 295 – 317: my main concerns relate to this section/the analysis of data under two frameworks (state-dependency and coping styles; see also comments above). The authors assess (and confirm) repeatability in fGC activity as the basis for being considered a “state” under the state-dependency framework. However, the authors don't follow through and do not test whether state (e.g. average fGCs; predictor variable) predicts personality expression (response

variable). Note that the first author's PhD thesis includes these exact analyses (<https://ethos.bl.uk/OrderDetails.do?uin=uk.bl.ethos.714481>). Instead, the authors test whether personality expression predicts fGCs. I suspect that the coping style framework was included to 'justify' this approach (the coping styles literature often focusses on correlations, 'ignoring' directionality/causality). Although, under both frameworks a correlation between GCs and personality expression should be present; the coping style framework focusses on links between on behavioural and physiological reactivity in challenging conditions ('coping with stressors'). I am not sure the present study/data set meets these (traditional) criteria, given the behavioural variables included and the time frames over which behavioural syndromes were assessed. The authors state (Lines 413-415) "our results suggest that proactive-reactive phenotypes....are expressed in non-manipulated settings" but the study was set out assuming this. Therefore, I recommend that the analyses should be presented as state (e.g. average fGCs; predictor variable) against personality expression (response variable).

- More generally, I wonder about 'Excitability' and the links to GCs. According to lines 420/421 Excitability includes both sociopositive and aggressive interactions for which we would predict opposite relationships with GCs. I know that the objective here is to link GCs to behavioural syndromes but grouping behavioural measures isn't always ideal when examining links to physiological stress levels. I think it is important to see how GC activity is linked to sociopositive/aggressive interactions (i.e. treating them separately) and this analysis should be conducted and included in the paper. This analysis may also give further insight into the observed sex difference in the relationship between Excitability and GCs (as aggression and/or affiliative behaviours are likely sex-dependent).

- Line 319: move to the beginning of the stats section

- The result section could be more concise and should include statistical test results.

- Have the authors considered combined models for males and females (and controlling for sex/fitting interactions)?

- Line 335/336: if GC reactivity is not repeatable, doesn't this rule out considering reactivity as a "state"; why should it explain personality expression?

- Lines 355ff: Given the suggested changes to data analysis, I don't provide detailed comments on the discussion/interpretation of results. However, the main issue regarding the two frameworks also exists in the discussion.

Line 390: change "wild" to natural"

Figure 2: please add real data points to effect plots

Decision letter (RSOS-190256.R0)

03-Apr-2019

Dear Dr Tkaczynski,

The editors assigned to your paper ("Repeatable glucocorticoid expression is associated with behavioural syndromes in males but not females in a wild primate") have now received

comments from reviewers. We would like you to revise your paper in accordance with the referee and Associate Editor suggestions which can be found below (not including confidential reports to the Editor). Please note this decision does not guarantee eventual acceptance.

Please submit a copy of your revised paper before 26-Apr-2019. Please note that the revision deadline will expire at 00.00am on this date. If we do not hear from you within this time then it will be assumed that the paper has been withdrawn. In exceptional circumstances, extensions may be possible if agreed with the Editorial Office in advance. We do not allow multiple rounds of revision so we urge you to make every effort to fully address all of the comments at this stage. If deemed necessary by the Editors, your manuscript will be sent back to one or more of the original reviewers for assessment. If the original reviewers are not available, we may invite new reviewers.

- Data accessibility

<http://datadryad.org/submit?journalID=RSOS&manu=RSOS-190256>

- Competing interests

- Authors' contributions

- Acknowledgements

- Funding statement

on behalf of Dr Alecia Carter (Associate Editor) and Kevin Padian (Subject Editor)
openscience@royalsociety.org

Associate Editor's comments (Dr Alecia Carter):

Associate Editor: 1

Comments to the Author:

Decision on RSOS-190256:

I have now received two constructive, convergent reviews of your manuscript, and have read it myself. Both reviewers agree that the manuscript is generally well-written and easy-to-follow, and the dataset is good for this kind of study. However, both reviewers raise some important points that should be addressed in a revision. I find myself in agreement with the reviewers' comments, both positive and negative, and had noted many of the points they raised in my own reading of the manuscript. In brief, the major points both reviewers highlight are that the authors do not and cannot measure stress reactivity using fGCs and that the direction of the tested correlation in the last set of analyses is incorrect/misleading. (Regarding this second point, I would encourage the authors to spend more time addressing the correlational nature of these data—it is possible, as the reviewers highlight, that behaviour is driving the fGC levels, as the models suggest, rather than fGCs driving behaviour, as the authors intended to analyse.)

In addition to the reviewers' very helpful feedback, I would also encourage the authors to address the following minor queries in a resubmission:

LL127-131: The authors describe 3 by-definition *uncorrelated* personality traits/syndromes (at L123), but then predict that all three will be determined by GC levels. Is this a realistic prediction? How can independent variables all be (predicted to be) affected in the same way and same direction by another latent variable? This would make all of them part of one syndrome, and functionally and statistically correlated, which is not in line with the authors' data. I suggest that the authors provide some nuance here – it is clear that there were no *a priori* predictions about which particular syndrome would be determined by GC levels (and an exploratory analysis is fine), but the authors should be clear that it cannot be all three of them.

Please make some mention of the small sample of individuals to variable ratio at some point. (The authors run analyses with 6-7 fixed effects on 12-13 individuals – that is only 2 independent data per variable. [I realise there is some disagreement about the “real” sample sizes of repeated-measures models, but this still needs to be acknowledged as a limitation on the generalisability of the results.]

Table 2: The long-term repeatability of GC activity seems to be driven by the one repeatable window (period 1) in both males and females. It seems to me that this pattern is overlooked in the Discussion? Please discuss the significance of this, and suggest a reason why this may be the case.

Comments to Author:

Reviewers' Comments to Author:

Reviewer: 1

Comments to the Author(s)

The paper explores very interesting topic and represents significant contribution to the field of animal personality. It is based on solid and intensive data collection. I think the paper is well written and the analysis clearly stated, I particularly appreciate the analysis of repeatability of stress levels measurement.

I have following comments:

37-39 Abstract: The last sentence is not clear to me, do you mean that behavioral syndromes are more associated with reproductive demands in females? You mean that their behavioral type changes based on reproductive stage? Or is it more that, in females, measures of physiological stress might be more associated with reproduction? I think the latter is rather the case.

In the abstract and elsewhere: I do not particularly like using the term “reactivity” for the second measure of stress. It is in fact variation in long-term levels. Traditionally the term reactivity refers to the difference between basal levels and peak levels provoked by stressor (as authors mention in the discussion). This is not what you are measuring in this study (and one can't do it with faecal samples). Although I understand your logic it is just not the correct use of the term and using the term for your measurement could be misleading to the readers. I would use the term variation instead.

Intro line 113: and Discussion. Here you mention one other study that explored stress levels and personality in baboons. This study is however not mentioned in the discussion. Given that it is presented as the only other study on the same topic, I would expect some discussion about its

results in comparison to this study (despite that it was only done on females it still should be mentioned)

line 149 and elsewhere: Given that the three study periods have different biological meaning (mating time vs post mating), I think it would be useful to give them names that include the meaning like mating period 1, post mating period 2. And use these in graphs too. Because when reader looks at graphs he/she should know what the time periods refers to (or put it to note)

line 242: Is Elo-rating particularly suitable for the subsequent analyses, or does it have some advantages compared to other more often used measures of rank e.g. rank order, I&IS method, Davids score? I am just curious if there are any advantages to use it in this particularly study.

line 280: specify if group identity was used as fixed or random effect (I assume it was fixed)

line 297: Why are the behavioral syndromes in the model as predictors/independent variables and the hormones as response/dependent variable? In a biological sense I would assume that it should be the other way round: the hormones are affecting the behavioral reaction/type (following same logic as e.g. genes are affecting behavior). I am aware that behavior can affect hormones, but I do not think this is the pattern you are trying to investigate. Also as implied in the introduction (lines 60-61: "endocrine functioning can have multi-modal effects on personality expression") following this I expected the hormones to be the independent variable. Is there any reason why authors selected this approach?

figure 2: Is it possible to plot the real data points to the graph too?

Reviewer: 2

Comments to the Author(s)

This paper investigates the link between behavioural syndromes and physiological stress levels (repeated faecal glucocorticoid metabolite measurements) in males and females of two groups of wild Barbary macaques (*Macaca sylvanus*). The main aim of the study is to test the "state-dependent" hypothesis which states that consistent individual differences in behaviour arise from differences in physiological state (e.g. hormones, metabolism). fGC activity (but not reactivity) was repeatable and the authors find evidence for a link between "Excitability" and fGCs in males but not females. Overall, this is an interesting data set collected under natural conditions. I do, however, have concerns/questions regarding the rationale of the conceptual framework(s) and associated statistical analysis which I outline below. I also provide more specific comments/suggestions.

- Line 28: delete "etc." in brackets

- Line 32/33: delete "a non-human primate species"

- Line 36: replace "; for females there were no....and personality measures" with "but not in females"

- Lines 37-39: The abstract ends on interpreting the non-significant result for females but doesn't say anything about the result for males and its implications

- Lines 72-75: "certain correlations of behaviour" – this is an inappropriate definition of coping styles (sensu Koolhaas et al., 1999, 2010)

- Lines 77/78: "as the behavioural syndrome concerned" is unclear

- Lines 80-82: please also include examples of studies finding evidence for repeatability in GCs
- Line 88-90: “both...state-dependent and coping style frameworks...” It is not clear why both concepts are used here and there is very little attempt to link these two frameworks and the predictions that can be derived. I think the manuscript would be stronger if the focus was solely on state-dependency (see comments Lines 295 – 317 below).
- Line 96: delete “now”
- Line 129/130: Are the predictions are that straightforward given that predictions for proactive/reactive individuals differ with regards to HPA axis activity/reactivity (see e.g. Koolhaas et al., 1999; Table 3)? I understand that reactivity here means variation and not necessarily high HPA axis activity in response to a stressor. This is likely to cause confusion especially amongst readers who are familiar with the coping style framework/literature.
- Lines 178ff: The description of reproductive states is insufficient/unclear. Why did you not use traditional categories (cycling, pregnant and lactating)? For periods 2 and 3 it should be possible to distinguish early versus late pregnancy (which can result in differences in fGCs, line 383) if birth dates are known. Similarly, if birth dates are known, conceptions could be estimated and females could be classified as cycling/early pregnant. The 4 categories for swellings are not defined and, in my opinion are not useful (and may unnecessarily affect model fitting).
- Lines 183-184: Delete sentence “In total, 2,616.....per subject”
- Lines 187ff: provide sample sizes (move lines 325-326 to methods section)
- Line 193: where samples transported on (dry) ice?
- Line 202: replace “removed” with “collected”
- Lines 202-204: I think this isn't necessary - say extracted following validated and published procedures and add citations?
- Line 207: add other non-human primates for which the assay has been used
- Lines 222ff: it is unclear if the quantification of behavioural syndromes was conducted specifically for the current study or if measures obtained for Reference 45 were “used” in the present study (unfortunately, I cannot access the paper).
- Lines 254-293: the entire section could be much clearer and concise. Also, what is the rationale for investigating repeatability short-and long-term with regards to subsequent analyses (line 259)?
- Lines 295 – 317: my main concerns relate to this section/the analysis of data under two frameworks (state-dependency and coping styles; see also comments above). The authors assess (and confirm) repeatability in fGC activity as the basis for being considered a “state” under the state-dependency framework. However, the authors don't follow through and do not test whether state (e.g. average fGCs; predictor variable) predicts personality expression (response variable). Note that the first author's PhD thesis includes these exact analyses (<https://ethos.bl.uk/OrderDetails.do?uin=uk.bl.ethos.714481>). Instead, the authors test whether personality expression predicts fGCs. I suspect that the coping style framework was included to ‘justify’ this approach (the coping styles literature often focusses on correlations, ‘ignoring’

directionality/causality). Although, under both frameworks a correlation between GCs and personality expression should be present; the coping style framework focusses on links between on behavioural and physiological reactivity in challenging conditions ('coping with stressors'). I am not sure the present study/data set meets these (traditional) criteria, given the behavioural variables included and the time frames over which behavioural syndromes were assessed. The authors state (Lines 413-415) "our results suggest that proactive-reactive phenotypes....are expressed in non-manipulated settings" but the study was set out assuming this. Therefore, I recommend that the analyses should be presented as state (e.g. average fGCs; predictor variable) against personality expression (response variable).

- More generally, I wonder about 'Excitability' and the links to GCs. According to lines 420/421 Excitability includes both sociopositive and aggressive interactions for which we would predict opposite relationships with GCs. I know that the objective here is to link GCs to behavioural syndromes but grouping behavioural measures isn't always ideal when examining links to physiological stress levels. I think it is important to see how GC activity is linked to sociopositive/aggressive interactions (i.e. treating them separately) and this analysis should be conducted and included in the paper. This analysis may also give further insight into the observed sex difference in the relationship between Excitability and GCs (as aggression and/or affiliative behaviours are likely sex-dependent).

- Line 319: move to the beginning of the stats section

- The result section could be more concise and should include statistical test results.

- Have the authors considered combined models for males and females (and controlling for sex/fitting interactions)?

- Line 335/336: if GC reactivity is not repeatable, doesn't this rule out considering reactivity as a "state"; why should it explain personality expression?

- Lines 355ff: Given the suggested changes to data analysis, I don't provide detailed comments on the discussion/interpretation of results. However, the main issue regarding the two frameworks also exists in the discussion.

Line 390: change "wild" to natural"

Figure 2: please add real data points to effect plots

Author's Response to Decision Letter for (RSOS-190256.R0)

See Appendix A.

RSOS-190256.R1 (Revision)

Review form: Reviewer 1

Is the manuscript scientifically sound in its present form?

No

Are the interpretations and conclusions justified by the results?

No

Is the language acceptable?

Yes

Is it clear how to access all supporting data?

Yes

Do you have any ethical concerns with this paper?

No

Have you any concerns about statistical analyses in this paper?

Yes

Recommendation?

Major revision is needed (please make suggestions in comments)

Comments to the Author(s)

I do appreciate the thorough revision and responses by the authors.

However I do not agree with their statistical approach and its justification. I am not professional statistician but I think switching response and predictor cannot be presented as "a statistically more conservative approach" to the same question. Because by switching the response and predictors you are asking completely different question not only biologically but also statistically. The model works with different standard errors and estimates different variability. So it is not "statistically more conservative alternative" to do it. Statistically the authors test something different compared to models where personality scores are response variable.

If you insists that the nature of the relationship is more of a correlation or that the direction is hard to establish, then you might consider other approaches to data analysis like e.g. major axis regression, which are more suitable for correlational data.

I do not consider multiple testing here to be a big issue. In any case the interpretation of the results should not be based only on significance level, but also on effect size. Commenting on effect size can illustrate for the reader if the affect is significant not only statistically but also biologically. E.g. if the predictor rises by xy units how much will the response change?

Also if you have only one predictor (hormonal levels) instead of the three personality predictors you might be able to use the sex as a predictor and decrease the number of models?

The new analysis where specific behaviors are put in the model as a predictors of hormonal levels is also questionable. I understand that it was a request from the second reviewer and this can bring interesting information. However the way this analysis is done could be seen statistically problematic. You first make a model using scores from PCA and then you make a model using the items from the PCA using the same response in both models. Moreover putting such high number of predictors in your model is not ideal given you sample size. Additionally one might assume that some of these behaviors are correlated (given that they cluster in PCA) are they? If yes, having them all in full model might lead to the non-significant results for all of them. This analysis might be considered as rather exploratory and so some sort of model selection approach

(adding or dropping variables) seems appropriate. Or even using some ordinary methods like RDA or CCA might provide some better insight into the relationship among behaviors and other variables including hormones.

There is a huge effort in the field behind this study, the paper is well written. I just think the data analysis approach can be better selected for the purpose of the study. Or at least better justified for the reader and the research question. It is very likely that the questions raised by the reviewers and editor will likely be asked by the readers e.g. Why are cortisol levels as a response (not the other way round)? Why are there separate models for each sex (why sex is not one of the predictors)? Whatever the final analysis strategy will be, the justification should be included in the data analysis section of the paper.

Review form: Reviewer 2

Is the manuscript scientifically sound in its present form?

No

Are the interpretations and conclusions justified by the results?

Yes

Is the language acceptable?

Yes

Is it clear how to access all supporting data?

Yes

Do you have any ethical concerns with this paper?

No

Have you any concerns about statistical analyses in this paper?

Yes

Recommendation?

Major revision is needed (please make suggestions in comments)

Comments to the Author(s)

Overall, most of my previous comments have been addressed. However, my main concern - fitting GCs as response and not as predictor variable, which was raised also by the other reviewer and the editor - has not been addressed. I am not convinced there is an issue 'regarding multiple testing' that couldn't be corrected for. The study is presented as a test of the state-hypothesis (e.g. lines 26-28; line 105, lines 118/119, lines 354-356) and the statistical approach does not reflect this.

Some additional comments:

Line 33: add brief detail explain "Excitability" in brackets

Lines 34-38: very vague/unclear sentences and no clear take-home message

Line 65: delete "including humans" (definition in line 52)

Lines 65-68: please cite primary literature for examples

Line 83 versus lines 116-117: contradictory; rephrase

Line 126-128: the way the prediction is formulated does not reflect the statistical analysis used; rephrase

Lines 127/128: use smaller case letters for excitable

Line 140: provide sample sizes (as stated in lines 146/147) here

Line 182: provide sample sizes (as stated in lines 190-191) here

Line 203: delete 'and ape species including'

Line 248: add "using the elo rating package in R [62]" after Elo-rating procedure [61] and delete the last sentence of the paragraph

Lines 342/343: I am still not convinced it is appropriate to run this model given that variation in GCs (former 'reactivity') was not repeatable. I don't think this analysis allows you "...to assess whether consistency in state is a pre-requisite..." In line 117 you state that repeatability is a requirement for the state-dependent model so - on paper - the rationale for including this analysis is not clear/justified

Line 381 and 385: "hormonal stress levels"

Line 453: what you mean by "stable stress levels"?

Decision letter (RSOS-190256.R1)

03-Jul-2019

Dear Dr Tkaczynski:

Manuscript ID RSOS-190256.R1 entitled "Repeatable glucocorticoid expression is associated with behavioural syndromes in males but not females in a wild primate" which you submitted to Royal Society Open Science, has been reviewed. The comments of the reviewer(s) are included at the bottom of this letter.

Please submit a copy of your revised paper before 26-Jul-2019. Please note that the revision deadline will expire at 00.00am on this date. If we do not hear from you within this time then it will be assumed that the paper has been withdrawn. In exceptional circumstances, extensions may be possible if agreed with the Editorial Office in advance. We do not allow multiple rounds of revision so we urge you to make every effort to fully address all of the comments at this stage. If deemed necessary by the Editors, your manuscript will be sent back to one or more of the original reviewers for assessment. If the original reviewers are not available we may invite new reviewers.

To revise your manuscript, log into <http://mc.manuscriptcentral.com/rsos> and enter your Author Centre, where you will find your manuscript title listed under "Manuscripts with Decisions." Under "Actions," click on "Create a Revision." Your manuscript number has been

appended to denote a revision. Revise your manuscript and upload a new version through your Author Centre.

- Ethics statement

- Data accessibility

- Competing interests

- Authors' contributions

- Acknowledgements

- Funding statement

on behalf of Dr Alecia Carter (Associate Editor) and Kevin Padian (Subject Editor)
openscience@royalsociety.org

Associate Editor Comments to Author (Dr Alecia Carter):

Dear authors,
Both of the original reviewers have now re-reviewed your manuscript, and although both reviewers highlight that this manuscript is well-written and the study well-executed, they both also still take issue with the central analysis. I also value the contribution that this manuscript could make, but I do agree with the reviewers that the analyses do not reflect your question and I disagree that it is statistically more conservative to analyse these data in one model when the model does not address the authors' question. Reviewer 1 provides good reasons for why this is statistically inappropriate.

I would really like to see this manuscript published in RSOS; however given the agreement between the reviewers' feedback and my own, I would encourage the authors to either (a) present the analyses as the reviewers have suggested, with the personality trait as the response variable and acknowledging the "multiple testing problem," or (b) present both analyses, which I hope would show the same relationship. I understand from the PhD thesis that this work is based on that the first author originally analysed the data using the method that the reviewers suggest--I hope that such a revision would thus be easy to implement and not take too much time.
Minor comment: L138: do the authors mean "2,000 m" here?

Subject Editor Comments to Author:

I agree with the AE's recommendation, and I am only logging a "major revision" decision so that you'll have a bit more time to consider revisions. This will not need to be seen again by external reviewers. Thanks very much!

Reviewer comments to Author:

Reviewer: 1

Comments to the Author(s)

I do appreciate the thorough revision and responses by the authors. However I do not agree with their statistical approach and its justification. I am not professional statistician but I think switching response and predictor cannot be presented as "a statistically more conservative approach" to the same question. Because by switching the response and

predictors you are asking completely different question not only biologically but also statistically. The model works with different standard errors and estimates different variability. So it is not “statistically more conservative alternative” to do it. Statistically the authors test something different compared to models where personality scores are response variable.

If you insists that the nature of the relationship is more of a correlation or that the direction is hard to establish, then you might consider other approaches to data analysis like e.g. major axis regression, which are more suitable for correlational data.

I do not consider multiple testing here to be a big issue. In any case the interpretation of the results should not be based only on significance level, but also on effect size. Commenting on effect size can illustrate for the reader if the affect is significant not only statistically but also biologically. E.g. if the predictor rises by xy units how much will the response change?

Also if you have only one predictor (hormonal levels) instead of the three personality predictors you might be able to use the sex as a predictor and decrease the number of models?

The new analysis where specific behaviors are put in the model as a predictors of hormonal levels is also questionable. I understand that it was a request from the second reviewer and this can bring interesting information. However the way this analysis is done could be seen statistically problematic. You first make a model using scores from PCA and then you make a model using the items from the PCA using the same response in both models. Moreover putting such high number of predictors in your model is not ideal given you sample size. Additionally one might assume that some of these behaviors are correlated (given that they cluster in PCA) are they? If yes, having them all in full model might lead to the non-significant results for all of them. This analysis might be considered as rather exploratory and so some sort of model selection approach (adding or dropping variables) seems appropriate. Or even using some ordinary methods like RDA or CCA might provide some better insight into the relationship among behaviors and other variables including hormones.

There is a huge effort in the field behind this study, the paper is well written. I just think the data analysis approach can be better selected for the purpose of the study. Or at least better justified for the reader and the research question. It is very likely that the questions raised by the reviewers and editor will likely be asked by the readers e.g. Why are cortisol levels as a response (not the other way round)? Why are there separate models for each sex (why sex is not one of the predictors)? Whatever the final analysis strategy will be, the justification should be included in the data analysis section of the paper.

Reviewer: 2

Comments to the Author(s)

Overall, most of my previous comments have been addressed. However, my main concern - fitting GCs as response and not as predictor variable, which was raised also by the other reviewer and the editor - has not been addressed. I am not convinced there is an issue ‘regarding multiple testing’ that couldn’t be corrected for. The study is presented as a test of the state-hypothesis (e.g. lines 26-28; line 105, lines 118/119, lines 354-356) and the statistical approach does not reflect this.

Some additional comments:

Line 33: add brief detail explain “Excitability” in brackets

Lines 34-38: very vague/unclear sentences and no clear take-home message

Line 65: delete “including humans” (definition in line 52)

Lines 65-68: please cite primary literature for examples

Line 83 versus lines 116-117: contradictory; rephrase

Line 126-128: the way the prediction is formulated does not reflect the statistical analysis used; rephrase

Lines 127/128: use smaller case letters for excitable

Line 140: provide sample sizes (as stated in lines 146/147) here

Line 182: provide sample sizes (as stated in lines 190-191) here

Line 203: delete 'and ape species including'

Line 248: add "using the elo rating package in R [62]" after Elo-rating procedure [61] and delete the last sentence of the paragraph

Lines 342/343: I am still not convinced it is appropriate to run this model given that variation in GCs (former 'reactivity') was not repeatable. I don't think this analysis allows you "...to assess whether consistency in state is a pre-requisite..." In line 117 you state that repeatability is a requirement for the state-dependent model so - on paper - the rationale for including this analysis is not clear/justified

Line 381 and 385: "hormonal stress levels"

Line 453: what you mean by "stable stress levels"?

Author's Response to Decision Letter for (RSOS-190256.R1)

See Appendix B.

Decision letter (RSOS-190256.R2)

08-Aug-2019

Dear Dr Tkaczynski,

I am pleased to inform you that your manuscript entitled "Repeatable glucocorticoid expression is associated with behavioural syndromes in males but not females in a wild primate" is now accepted for publication in Royal Society Open Science.

Kind regards,

on behalf of Dr Alecia Carter (Associate Editor) and Kevin Padian (Subject Editor)
openscience@royalsociety.org

Appendix A

Associate Editor's comments (Dr Alecia Carter):

Associate Editor: 1

Comments to the Author:

Decision on RSOS-190256:

I have now received two constructive, convergent reviews of your manuscript, and have read it myself. Both reviewers agree that the manuscript is generally well-written and easy-to-follow, and the dataset is good for this kind of study. However, both reviewers raise some important points that should be addressed in a revision. I find myself in agreement with the reviewers' comments, both positive and negative, and had noted many of the points they raised in my own reading of the manuscript. In brief, the major points both reviewers highlight are that the authors do not and cannot measure stress reactivity using fGCs and that the direction of the tested correlation in the last set of analyses is incorrect/misleading. (Regarding this second point, I would encourage the authors to spend more time addressing the correlational nature of these data—it is possible, as the reviewers highlight, that behaviour is driving the fGC levels, as the models suggest, rather than fGCs driving behaviour, as the authors intended to analyse.)

We thank the editor and reviewers for their detailed feedback. Regarding our measure of “reactivity”, we have now rephrased this measure as an overall *variation* in hormonal stress (e.g. lines 32, 117, 210-211), and have highlighted the limitation of fGCs for measuring any form of reactivity (lines 402-406).

Regarding the second point and the correlational nature of the study, we dedicate more of the discussion to this point, highlighting, as stated by the reviewers as well, that the behaviours could be driving the stress patterns rather than vice versa (lines 448-454). However, we do not fully agree with the reviewers that the best way to address this would be to run models with each syndrome as the dependent variable as this would lead to multiple testing. Running a single model per sex per stress variable seems the most parsimonious and statistically conservative approach. The ideal way to test state dependence would be to test how changes in state relate to change in behaviour. We expand on this in the Discussion (lines 455-465).

In addition to the reviewers' very helpful feedback, I would also encourage the authors to address the following minor queries in a resubmission:

LL127-131: The authors describe 3 by-definition **uncorrelated** personality traits/syndromes (at L123), but then predict that all three will be determined by GC levels. Is this a realistic prediction? How can independent variables all be (predicted to be) affected in the same way and same direction by another latent variable? This would make all of them part of one syndrome, and functionally and statistically correlated, which is not in line with the authors' data. I suggest that the authors provide some nuance here—it is clear that there were no *a priori* predictions about which particular syndrome would be determined by GC levels (and an exploratory analysis is fine), but the authors should be clear that it cannot be all three of them.

This is a fair criticism, although we did originally predict that Excitability should correlate with physiological stress given its structural similarity to other personality/syndromes which correlate with stress physiological in other studies (e.g. Koolhaas et al, 1999, 2010). Therefore, we have made a specific prediction for this behavioural syndrome while highlighting the difficulty of making *a priori* predictions for the Tactility and Sociability (lines 129-133).

Please make some mention of the small sample of individuals to variable ratio at some point. (The authors run analyses with 6-7 fixed effects on 12-13 individuals—that is only 2 independent data per

variable. [I realise there is some disagreement about the “real” sample sizes of repeated-measures models, but this still needs to be acknowledged as a limitation on the generalisability of the results.]

We now acknowledge our small sample size in lines 456-458 of the Discussion. In addition to this we have provided further details on model stability (robustness to random effects being dropped from the model) in the Methods section in lines 320-323.

Table 2: The long-term repeatability of GC activity seems to be driven by the one repeatable window (period 1) in both males and females. It seems to me that this pattern is overlooked in the Discussion? Please discuss the significance of this, and suggest a reason why this may be the case.

This is likely to be the case given the low repeatability coefficients during the other periods. We expand on this and its implications for the time frames in which to measure future repeatability estimates in the Discussion (lines 377-386).

Reviewers' Comments to Author:

Reviewer: 1

Comments to the Author(s)

The paper explores very interesting topic and represents significant contribution to the field of animal personality. It is based on solid and intensive data collection. I think the paper is well written and the analysis clearly stated, I particularly appreciate the analysis of repeatability of stress levels measurement.

I have following comments:

37-39 Abstract: The last sentence is not clear to me, do you mean that behavioral syndromes are more associated with reproductive demands in females? You mean that their behavioral type changes based on reproductive stage? Or is it more that, in females, measures of physiological stress might be more associated with reproduction? I think the latter is rather the case.

Following this comment and others made by the other reviewer and editor, the abstract has been largely re-written. We agree that the latter is the case and have now made this clearer in the new abstract (lines 34-38).

In the abstract and elsewhere: I do not particularly like using the term “reactivity” for the second measure of stress. It is in fact variation in long-term levels. Traditionally the term reactivity refers to the difference between basal levels and peak levels provoked by stressor (as authors mention in the discussion). This is not what you are measuring in this study (and one can't do it with faecal samples). Although I understand your logic it is just not the correct use of the term and using the term for your measurement could be misleading to the readers. I would use the term variation instead.

This is a valid criticism and we have now renamed this particular measure as an indicator of overall variation in hormonal stress (e.g. lines 32, 117, 208-211). Further, we have highlighted the limitation of faecal glucocorticoids for measuring any form of reactivity in our Discussion (lines 402-406).

Intro line 113: and Discussion. Here you mention one other study that explored stress levels and personality in baboons. This study is however not mentioned in the discussion. Given that it is presented as the only other study on the same topic, I would expect some discussion about its results in comparison to this study (despite that it was only done on females it still should be mentioned)

We now discuss the results of the Seyfarth et al's (2012) paper in light of our own results in the Discussion. In particular, we examine why this study was able to establish a relationship between female behavioural syndromes and hormonal stress yet we were unable to do the same (lines 485-496).

line 149 and elsewhere: Given that the three study periods have different biological meaning (mating time vs post mating), I think it would be useful to give them names that include the meaning like mating period 1, post mating period 2. And use these in graphs too. Because when reader looks at graphs he/she should know what the time periods refers to (or put it to note)

We have made this correction and refer to the time periods in terms of their biological meaning throughout the ms. The table and figure captions, and text on tables, have also been relabelled accordingly.

line 242: Is Elo-rating particularly suitable for the subsequent analyses, or does it have some advantages compared to other more often used measures of rank e.g. rank order, I&S method, Davids score? I am just curious if there are any advantages to use it in this particularly study.

During this study, there were frequent rank changes in both groups, including in the females in one of the groups (Green group), which is unusual for a species with a normally stable female hierarchy (Fooden, 2007). Given the relative frequency of these changes, the iterative rank measure provided by Elo-ratings seemed appropriate. The scores provided by this approach are also useful for measuring the magnitude of difference between two individuals: for example, the top-ranking male in one group may be particularly despotic and aggressive compared to a male in another group where dominance interactions are less frequent. For the former, the top-ranking male would have a score much greater than the next highest ranking, whereas in the latter, the top-ranking male may have a rank score quite close to his nearest rival.

The associations between rank and hormonal stress are not consistent in the primate literature (e.g. Abbott et al, 2003), which is possibly not only a consequence of societal differences between species, but also maybe the range of measures to determine rank. Nevertheless, former studies utilising David's scores in this population of macaques also failed to find associations between rank and hormonal stress (Young et al, 2014). Therefore, we feel that the absence of an association between rank and hormonal stress observed in our study is unlikely to be the result of

using an inappropriate rank measure and more likely reflects the actual lack of a consistent relationship between these factors in this species.

line 280: specify if group identity was used as fixed or random effect (I assume it was fixed)

Group was included as a fixed effect in all models (lines 280-281).

line 297: Why are the behavioral syndromes in the model as predictors/independent variables and the hormones as response/dependent variable? In a biological sense I would assume that it should be the other way round: the hormones are affecting the behavioral reaction/type (following same logic as e.g. genes are affecting behavior). I am aware that behavior can affect hormones, but I do not think this is the pattern you are trying to investigate. Also as implied in the introduction (lines 60-61: "endocrine functioning can have multi-modal effects on personality expression") following this I expected the hormones to be the independent variable. Is there any reason why authors selected this approach?

As advantageous as the suggested approach would be from a point of presentation (in terms of the state resulting in the behaviour), it would be problematic statistically as it would lead to multiple models which would essentially be testing the same thing (it would mean six models, three per sex). All these models would essentially test the same hypothesis, as the study is correlational. Although we propose that the state of stress physiology is leading to the stable behavioural patterns observed, we cannot rule out that the reverse may be occurring (stable behaviour leading to stable stress). In light of your comments, we may have originally overstated the directionality of the relationship, which cannot be determined beyond doubt. In the revised manuscript we now discuss this in greater detail and also suggest a way of how best to capture the directionality of the relationship in future studies (455-465).

figure 2: Is it possible to plot the real data points to the graph too?

These have been added (line 832).

Reviewer: 2

Comments to the Author(s)

This paper investigates the link between behavioural syndromes and physiological stress levels (repeated faecal glucocorticoid metabolite measurements) in males and females of two groups of wild Barbary macaques (*Macaca sylvanus*). The main aim of the study is to test the "state-dependent" hypothesis which states that consistent individual differences in behaviour arise from differences in physiological state (e.g. hormones, metabolism). fGC activity (but not reactivity) was repeatable and the authors find evidence for a link between "Excitability" and fGCs in males but not females. Overall, this is an interesting data set collected under natural conditions. I do, however, have concerns/questions regarding the rationale of the conceptual framework(s) and associated statistical analysis which I outline below. I also provide more specific comments/suggestions.

- Line 28: delete “etc.” in brackets

This has been done.

- Line 32/33: delete “a non-human primate species”

This has been done

- Line 36: replace “; for females there were no....and personality measures” with “but not in females”

This has been corrected

- Lines 37-39: The abstract ends on interpreting the non-significant result for females but doesn't say anything about the result for males and its implications

We have reworked the abstract and framed the concluding sentence regarding the sex differences in relation to the state dependent model (lines 34-38).

- Lines 72-75: “certain correlations of behaviour” – this is an inappropriate definition of coping styles (sensu Koolhaas et al., 1999, 2010)

We have changed the terminology to “patterns of behaviour”

- Lines 77/78: “as the behavioural syndrome concerned” is unclear

This has been changed to “An assumption of the state-dependent hypothesis is that a behavioural syndrome-associated state, such as GC measures, should be as consistent and repeatable as the behavioural syndrome to which it is associated”.

- Lines 80-82: please also include examples of studies finding evidence for repeatability in GCs

These examples are GC studies, which has been made clear now in the text (lines 79-81).

- Line 88-90: “both...state-dependent and coping style frameworks...” It is not clear why both concepts are used here and there is very little attempt to link these two frameworks and the predictions that can be derived. I think the manuscript would be stronger if the focus was solely on state-dependency (see comments Lines 295 – 317 below).

We comment more directly on this issue under the comments regarding lines 295-317.

- Line 96: delete “now”

This has been done.

- Line 129/130: Are the predictions are that straightforward given that predictions for proactive/reactive individuals differ with regards to HPA axis activity/reactivity (see e.g. Koolhaas et al., 1999; Table 3)? I understand that reactivity here means variation and not necessarily high HPA axis activity in response to a stressor. This is likely to cause confusion especially amongst readers who are familiar with the coping style framework/literature.

As we have changed our terminology regarding the GC reactivity variable to GC variation, we feel the prediction is now more in keeping with the proactive-reactive literature. We have also revised the predictions to make it clear that as we expected the strongest associations between Excitability and physiological stress, and as all the behavioural syndromes are independent of each other, that we were not able to make *a priori* predictions for the other two behavioural syndromes.

- Lines 178ff: The description of reproductive states is insufficient/unclear. Why did you not use traditional categories (cycling, pregnant and lactating)? For periods 2 and 3 it should be possible to distinguish early versus late pregnancy (which can result in differences in fGCs, line 383) if birth dates are known. Similarly, if birth dates are known, conceptions could be estimated and females could be classified as cycling/early pregnant. The 4 categories for swellings are not defined and, in my opinion are not useful (and may unnecessarily affect model fitting).

We adopted reproductive state categories previously used and published in Barbary macaque research (Young et al, 2013). Young et al (2013) were able to show that swellings in female Barbary macaques are honest signals of ovulation, but we concede that in macaques, no clear associations between maximal swelling and glucocorticoids have been established.

We compared a model containing these sex swelling categories to one without and found that the model without the sexual swelling variable had a lower AIC score (sex state model 3031.3 vs no sex state model 3026.7), indicative that the sex swelling variable did not better explain the data despite the increase in model complexity. Therefore, in the new ms, we have removed the sexual swelling variable and references to it entirely from the analysis.

Unfortunately, exact birth dates are not known: the macaques were infrequently followed after the end of our study, but other researchers were able to validate that all females subsequently gave birth between April and May in both field seasons. Therefore, we are unable to distinguish precisely between early and late stage pregnancy in this study. Indeed, even with precise birth dates, without an daily assessment of GC patterns over pregnancy coupled with other sex hormone data, distinguishing between “early” and “late” would be difficult. Thus, we leave time period as conflated with pregnancy in our analyses and justify this in lines 275-278.

- Lines 183-184: Delete sentence "In total, 2,616.....per subject"

This has been removed

- Lines 187ff: provide sample sizes (move lines 325-326 to methods section)

The faecal sample sizes have been moved to line 190-191.

- Line 193: where samples transported on (dry) ice?

The method of transportation has been clarified on line 187-189.

- Line 202: replace "removed" with "collected"

This has been changed.

- Lines 202-204: I think this isn't necessary - say extracted following validated and published procedures and add citations?

This has been done (line 197).

- Line 207: add other non-human primates for which the assay has been used

This assay was concurrently validated for Barbary macaques, long-tailed macaques, chimpanzees and gorillas (Heistermann et al, 2006), but we also include references for other macaque species in lines 202-204.

- Lines 222ff: it is unclear if the quantification of behavioural syndromes was conducted specifically for the current study or if measures obtained for Reference 45 were "used" in the present study (unfortunately, I cannot access the paper).

The present study and the study resulting in reference 45 were conducted concurrently. As the results of the behavioural syndrome *quantification* are now published, we reference that here. This has been clarified in lines 223-226. We anticipate Tkaczynski et al, (2018; i.e. ref 45) will be more widely accessible once the paper moves from advance online status to being formally published in the next volume of the journal although you may be able to access it from the University of Roehampton's repository:

[https://pure.roehampton.ac.uk/portal/en/publications/measuring-personality-in-the-field-an-in-situ-comparison-of-personality-quantification-methods-in-wild-barbary-macaques-macaca-sylvanus\(beaa7ac3-c246-4d99-9a53-2e963d3b8184\).html](https://pure.roehampton.ac.uk/portal/en/publications/measuring-personality-in-the-field-an-in-situ-comparison-of-personality-quantification-methods-in-wild-barbary-macaques-macaca-sylvanus(beaa7ac3-c246-4d99-9a53-2e963d3b8184).html)).

- Lines 254-293: the entire section could be much clearer and concise. Also, what is the rationale for investigating repeatability short-and long-term with regards to subsequent analyses (line 259)?

We have simplified our description of the analyses performed. The short-term and long-term repeatability was performed to see if different social/environmental settings in anyway affected the repeatability of fGCs, especially given pregnancy is associated with elevated GCs. We discuss in more detail the implications of short-term repeatability only being evident during the mating seasons and whether this is driving the overall repeatability in fGCs observed (lines 371-386).

- Lines 295 – 317: my main concerns relate to this section/the analysis of data under two frameworks (state-dependency and coping styles; see also comments above). The authors assess (and confirm) repeatability in fGC activity as the basis for being considered a “state” under the state-dependency framework. However, the authors don’t follow through and do not test whether state (e.g. average fGCs; predictor variable) predicts personality expression (response variable). Note that the first author’s PhD thesis includes these exact analyses (<https://ethos.bl.uk/OrderDetails.do?uin=uk.bl.ethos.714481>). Instead, the authors test whether personality expression predicts fGCs. I suspect that the coping style framework was included to ‘justify’ this approach (the coping styles literature often focusses on correlations, ‘ignoring’ directionality/causality). Although, under both frameworks a correlation between GCs and personality expression should be present; the coping style framework focusses on links between on behavioural and physiological reactivity in challenging conditions (‘coping with stressors’). I am not sure the present study/data set meets these (traditional) criteria, given the behavioural variables included and the time frames over which behavioural syndromes were assessed. The authors state (Lines 413-415) “our results suggest that proactive-reactive phenotypes...are expressed in non-manipulated settings” but the study was set out assuming this. Therefore, I recommend that the analyses should be presented as state (e.g. average fGCs; predictor variable) against personality expression (response variable).

We thank the reviewer for their comments on this component of the manuscript. The reason for the model structures presented (as opposed to those in the PhD thesis where this work originates from) was a concern regarding multiple testing (a concern not properly addressed in the aforementioned thesis). As you point out, our study is correlational and unable to categorically determine the direction of the relationship between hormonal stress and behavioural syndromes. However, to include the behavioural syndromes as the dependent variables would result in three times the number of models run and in increased aggregation of data, something we feel is best avoided. One option might have been using a MANOVA with a multiple response output (each behavioural syndrome score as a potential output), but we are not aware of a mixed effect model alternative to this.

Our original manuscript may have conflated the coping style and state-dependent frameworks, and we have now edited the manuscript to clarify this. While coping styles have traditionally been tested in experiments, either in captivity or via temporary trapping of captive populations, with such experiments presenting direct stressors and measuring immediate responses, the theory does not oppose that coping styles behaviours might manifest consistently in normal social situations. Furthermore, the variation in coping style to every day stressors is proposed as a driver of evolutionary processes: “Individual variation in coping with challenges encountered in the

natural habitat determines evolutionary fitness and is considered to be the prerequisite of speciation” (Koolhaas et al., 2010). Given the Excitability behavioural syndrome’s structural similarity to the proactive-reactive axis proposed in the coping style literature, we feel it still is necessary to discuss our results, at least in part, in relation to this literature and framework. However, we now point out in the Discussion (lines 427-434) that coping styles are usually considered in terms of immediate behavioural and physiological responses to direct stressors – something we – unfortunately - cannot test for with our dataset.

- More generally, I wonder about ‘Excitability’ and the links to GCs. According to lines 420/421 Excitability includes both sociopositive and aggressive interactions for which we would predict opposite relationships with GCs. I know that the objective here is to link GCs to behavioural syndromes but grouping behavioural measures isn’t always ideal when examining links to physiological stress levels. I think it is important to see how GC activity is linked to sociopositive/aggressive interactions (i.e. treating them separately) and this analysis should be conducted and included in the paper. This analysis may also give further insight into the observed sex difference in the relationship between Excitability and GCs (as aggression and/or affiliative behaviours are likely sex-dependent).

Both the state-dependent and coping style frameworks assume that an underlying mechanism leads to a consistent correlation between a particular set of behaviours. In this sense, it is the linkage between these behaviours that is more important than the constituent behaviours themselves. Nevertheless, we agree with the reviewer that an important *post hoc* analysis should be determining if any one constituent behaviour within a behavioural syndrome is driving the correlation with stress physiology. Therefore, we present a new analysis in the revised manuscript in which we find no association between GC activity and the constituent behaviours of Excitability (lines 312-318; lines 337-340; lines 440-442; Table 4).

- Line 319: move to the beginning of the stats section

This has been done.

- The result section could be more concise and should include statistical test results.

The entire text of the Results section has been shortened and now provides the tests and p values for results of interest (lines 325-351).

- Have the authors considered combined models for males and females (and controlling for sex/fitting interactions)?

We did indeed consider this but when combining the models our sample size was insufficient for including sex as factor and its interactions with the other variables.

- Line 335/336: if GC reactivity is not repeatable, doesn't this rule out considering reactivity as a "state"; why should it explain personality expression?

This is a valid criticism and we did consider leaving these models out once we failed to establish repeatability for the reactivity measure (now called measure of variation rather than reactivity following comments from the other reviewers). However, by including this model we were able to assess whether consistency in state is a pre-requisite for state-behavioural syndrome associations. The absence of a relationship between stress variation and any behavioural syndrome lends some weight to the main result, i.e. that a repeatable state (stress activity) was associated with repeatable and correlated behaviour (Excitability).

- Lines 355ff: Given the suggested changes to data analysis, I don't provide detailed comments on the discussion/interpretation of results. However, the main issue regarding the two frameworks also exists in the discussion.

Line 390: change "wild" to natural"

This has been done.

Figure 2: please add real data points to effect plots

These have been added.

Appendix B

Dr Patrick Tkaczynski
Max Planck Institute for
Evolutionary Anthropology
Dept of Primatology
Deutscher Platz 6
Leipzig
Germany
patrick.tkaczynski@eva.mpg.de

RE: Manuscript revision
24th July 2019

Dear Editorial Office,

Please accept this covering letter supporting the submission of a further revision to our manuscript titled: "*Repeatable glucocorticoid expression is associated with behavioural syndromes in males but not females in a wild primate*", again submitted for exclusive consideration of publication as an article in *Royal Society Open Science*.

The paper provides the first examination of sex-specific associations between hormonal stress and behavioural syndromes using non-experimental behavioural syndrome quantification and non-invasive hormonal sampling. Using wild Barbary macaques (*Macaca sylvanus*) as our study species, we found consistent inter-individual differences in glucocorticoid expression in both sexes, however, correlations between this glucocorticoid expression and behavioural syndromes were only observed in males and not females. We consider these results principally in the context of the state-dependent model for understanding how and why behavioural syndromes evolve and are maintained in a population – a popular topic currently within behavioural ecology.

We have incorporated comments from the handling editor and the two reviewers, which we feel have significantly improved the clarity and quality of the manuscript. Specifically, we have changed the analyses examining relationships between behavioural syndromes and glucocorticoid measures to ensure the former is now the dependent variable, as requested by the editor and reviewers.

We are grateful for their advice and the opportunity to present this revision. Our specific responses to the both the editor and reviewers are attached below to this letter.

Please address all future correspondence to myself (patrick.tkaczynski@eva.mpg.de; pjtresearchltd@gmail.com).

Yours sincerely,

Dr Patrick Tkaczynski

Associate Editor Comments to Author (Dr Alecia Carter):

Dear authors,

Both of the original reviewers have now re-reviewed your manuscript, and although both reviewers highlight that this manuscript is well-written and the study well-executed, they both also still take issue with the central analysis. I also value the contribution that this manuscript could make, but I do agree with the reviewers that the analyses do not reflect your question and I disagree that it is statistically more conservative to analyse these data in one model when the model does not address the authors' question. Reviewer 1 provides good reasons for why this is statistically inappropriate.

I would really like to see this manuscript published in RSOS; however given the agreement between the reviewers' feedback and my own, I would encourage the authors to either (a) present the analyses as the reviewers have suggested, with the personality trait as the response variable and acknowledging the "multiple testing problem," or (b) present both analyses, which I hope would show the same relationship. I understand from the PhD thesis that this work is based on that the first author originally analysed the data using the method that the reviewers suggest--I hope that such a revision would thus be easy to implement and not take too much time.

Minor comment: L138: do the authors mean "2,000 m" here?

Yes, this has been corrected (line 139).

Subject Editor Comments to Author:

I agree with the AE's recommendation, and I am only logging a "major revision" decision so that you'll have a bit more time to consider revisions. This will not need to be seen again by external reviewers. Thanks very much!

Reviewer comments to Author:

Reviewer: 1

Comments to the Author(s)

I do appreciate the thorough revision and responses by the authors.

However I do not agree with their statistical approach and its justification. I am not professional statistician but I think switching response and predictor cannot be presented as "a statistically more conservative approach" to the same question. Because by switching the response and predictors you are asking completely different question not only biologically but also statistically. The model works with different standard errors and estimates different variability. So it is not "statistically more conservative alternative" to do it. Statistically the authors test something different compared to models where personality scores are response variable.

If you insists that the nature of the relationship is more of a correlation or that the direction is hard to establish, then you might consider other approaches to data analysis like e.g. major axis regression, which are more suitable for correlational data.

I do not consider multiple testing here to be a big issue. In any case the interpretation of the results should not be based only on significance level, but also on effect size. Commenting on effect size can illustrate for the reader if the affect is significant not only statistically but also biologically. E.g. if the predictor rises by xy units how much will the response change?

Also if you have only one predictor (hormonal levels) instead of the three personality predictors you might be able to use the sex as a predictor and decrease the number of models?

We have adjusted the critiqued analyses largely as suggested, using the suggested approaches as appropriate to the dataset and question. Because we could not simply reverse the predictor and dependent variable (as the reviewer points out, the model structure needs to change, since we had multiple measures of GC over the study period), we used the random effect estimates from the GC repeatability models as our predictor for behavioural syndrome scores (method described lines 304-312; results 350-360; Pinheiro & Bates [2000; see main manuscript for full reference]). These estimates reflect the GC phenotype for an individual, i.e. their average GC level within a particular time period given the fixed effects of temperature, group and rank within that given time period.

In our original approach, we used the GC-behavioural syndrome models (with GCs as the dependent variable) to assess how control variables (monthly temperature, group identity etc.) affect GC activity and variation. Now, as this part of the manuscript uses behavioural syndromes as the dependent variable, we instead test how the aforementioned control variables affect GC activity and variation by using a full model approach on the repeatability models for GC activity and variation (method described lines 295-297; results 335-348). In this way, we are able to still discuss factors affecting variability in GC measures and their repeatability, prior to examining the relationship between GC measures and behavioural syndromes.

The new analysis where specific behaviors are put in the model as a predictors of hormonal levels is also questionable. I understand that it was a request from the second reviewer and this can bring interesting information. However the way this analysis is done could be seen statistically problematic. You first make a model using scores from PCA and then you make a model using the items from the PCA using the same response in both models. Moreover putting such high number of predictors in your model is not ideal given you sample size. Additionally one might assume that some of these behaviors are correlated (given that they cluster in PCA) are they? If yes, having them all in full model might lead to the non-significant results for all of them. This analysis might be considered as rather exploratory and so some sort of model selection approach (adding or dropping variables) seems appropriate. Or even using some ordinary methods like RDA or CCA might provide some better insight into the relationship among behaviors and other variables including hormones.

We reanalysed these potential relationships with a series of linear models using the behaviour frequency as the dependent variable and the GC phenotype variable as the predictor (method described lines 313-321; results 352-355). Although this resulted in a number of additional models this approach did provide largely similar results to those reported previously suggesting that the observed relationship between Excitability and GC activity in male macaques was not driven by one particular constituent behaviour (discussed line 451-460).

There is a huge effort in the field behind this study, the paper is well written. I just think the data analysis approach can be better selected for the purpose of the study. Or at least better justified for the reader and the research question. It is very likely that the questions raised by the reviewers and editor will likely be asked by the readers e.g. Why are cortisol levels as a response (not the other way round)? Why are there separate models for each sex (why sex is not one of the predictors)? Whatever the final analysis strategy will be, the justification should be included in the data analysis section of the paper.

We thank the reviewer for their feedback and we have now addressed these issues with the data analyses, both in how they were conducted and then presented in the manuscript.

Reviewer: 2

Comments to the Author(s)

Overall, most of my previous comments have been addressed. However, my main concern - fitting GCs as response and not as predictor variable, which was raised also by the other reviewer and the editor – has not been addressed. I am not convinced there is an issue ‘regarding multiple testing’ that couldn’t be corrected for. The study is presented as a test of the state-hypothesis (e.g. lines 26-28; line 105, lines 118/119, lines 354-356) and the statistical approach does not reflect this.

As per our comments to the first reviewer, we now provide a new analysis using the GC measure as a predictor of behavioural syndrome scores. The results (significance and direction of relationships) remain essentially the same and we agree this analysis and presentation of results should be more satisfying to the reader. Clarification on the methods used (lines 304-312) and their results (lines 350-360) have been added to the manuscript. Please see our comments to reviewer 1 for further details on the approach used.

Some additional comments:

Line 33: add brief detail explain “Excitability” in brackets

This has been added (line 33-34).

Lines 34-38: very vague/unclear sentences and no clear take-home message

We have re-written these sentences. We now highlight that given the absence of a state-behavioural syndrome association in females in our study, such state-behavioural syndrome associations are not be generalizable within a species and that future research needs to incorporate data on sex differences in order to understand the emergence and maintenance of behavioural syndromes (lines 35-39).

Line 65: delete “including humans” (definition in line 52)

This has been done.

Lines 65-68: please cite primary literature for examples

We now cite primary literature here and have also included general reviews of the HPA and GC role in stress response (line 66 and line 69).

Line 83 versus lines 116-117: contradictory; rephrase

We have rephrased this and included an additional definition of repeatability in GC measures. We also explain in greater detail how this relates to the state-dependent framework. This clarifies the apparent contradiction (lines 120-121).

Line 126-128: the way the prediction is formulated does not reflect the statistical analysis used; rephrase

The statistical approach has now been changed and closely tracks the formulated predictions (see previous comments).

Lines 127/128: use smaller case letters for excitable

This has been changed.

Line 140: provide sample sizes (as stated in lines 146/147) here

This has been changed.

Line 182: provide sample sizes (as stated in lines 190-191) here

This has been changed.

Line 203: delete 'and ape species including'

This has been deleted.

Line 248: add "using the elo rating package in R [62]" after Elo-rating procedure [61] and delete the last sentence of the paragraph

This has been done.

Lines 342/343: I am still not convinced it is appropriate to run this model given that variation in GCs (former 'reactivity') was not repeatable. I don't think this analysis allows you "...to assess whether consistency in state is a pre-requisite..." In line 117 you state that repeatability is a requirement for the state-dependent model so – on paper – the rationale for including this analysis is not clear/justified

We have now removed this analysis. Instead, we now state that these models could not be run because the GC measure were not repeatable (lines 360-361).

Line 381 and 385: "hormonal stress levels"

This has been corrected.

Line 453: what you mean by "stable stress levels"?

This has been clarified to mean "consistent inter-individual differences".